# Adaptive particle representation of fluorescence microscopy images

Bevan L. Cheeseman [1,2], Ulrik Günther [1,2], Krzysztof Gonciarz [1,2], Mateusz Susik[1,2] & Ivo F. Sbalzarini [1,2]

Modern microscopes create a data deluge with gigabytes of data generated each second, and terabytes per day. Storing and processing this data is a severe bottleneck, not fully alleviated by data compression. We argue that this is because images are processed as grids of pixels. To address this, we propose a content-adaptive representation of fluorescence microscopy images, the Adaptive Particle Representation (APR). The APR replaces pixels with particles positioned according to image content. The APR overcomes storage bottlenecks, as data compression does, but additionally overcomes memory and processing bottlenecks. Using noisy 3D images, we show that the APR adaptively represents the content of an image while maintaining image quality and that it enables orders of magnitude benefits across a range of image processing tasks. The APR provides a simple and efficient content-aware representation of fluosrescence microscopy images.

[1] Chair of Scientific Computing for Systems Biology, Faculty of Computer Science, TU Dresden, 01069 Dresden, Germany. [2] Center for Systems Biology Dresden, Max Planck Institute of Molecular Cell Biology and Genetics, Pfotenhauerstr. 108, 01307 Dresden, Germany. Correspondence and requests for materials should be addressed to I.F.S. (email: ivos@mpi-cbg.de)

Developments in fluorescence microscopy[1–3], labeling chemistry[4], and genetics[5] provide the potential to capture and track biological structures at high resolution in both space and time. Such data are vital for understanding spatio-temporal processes in biology[6]. Unfortunately, fluorescence microscopes do not directly output the shapes and locations of objects through time. Instead, they produce raw data, potentially terabytes of 3D images[7], from which the desired spatiotemporal information must be extracted by image processing. Handling the large image data and extracting information from the raw microscopy images currently presents the main bottleneck[7–9]. We propose that at the core of the problem is not the amount of information contained in the images, but how the information is encoded—usually as a uniform grid of pixels. While data compression can alleviate storage issues, it does not reduce memory usage nor computational cost as all processing must still be done on the original, uncompressed data.

Processing bottlenecks are effectively avoided by the human visual system, which solves a similar problem of inferring object shapes and locations from photon counts. In part, the human visual system achieves this by adaptively sampling the scene depending on its content[10], while adjusting to the dynamic range of intensity variations[11]. This adaptive sampling works by selectively focusing the attention of the eyes on areas with potentially high information content[10]. This selective focus then enables the efficient inference of information about the scene at a high effective resolution by directing the processing capacity of the retina and the visual cortex. As in fluorescence microscopy, the information in different areas of a scene is not encoded in absolute intensity differences, but in relative differences compared to the local brightness. The human visual system maintains effective adaptive sampling across up to nine orders of magnitude of brightness[11] by using local gain control mechanisms that adjust to, and account for, changes in the dynamic range of intensity variations. Together, adaptation and local gain control enable the visual system to provide a high rate of information content using as little as $1\mathrm{MB\,s}^{-1}$ of data from the retina[12]. In contrast, the information-to-data ratio in pixel representations of fluorescence microscopy images is much lower and is governed by the spatial and temporal resolution of the images rather than by their contents.

In light of this, an ideal representation of fluorescence microscopy images would share the features of adaptation and local gain control with the human visual system. We posit that any image representation aiming to achieve this should fulfill the following representation criteria (RC):

RC1: It must guarantee a user-controllable representation accuracy for noise-free images and must not reduce the signal-to-noise ratio of noisy images.

RC2: Memory and computational cost of the representation must be proportional to the information content of an image, and not to its number of pixels.

RC3: It must be possible to rapidly convert a given pixel image into that representation with a computational cost at most proportional to the number of input pixels.

RC4: The representation must reduce both computational cost and memory cost of image-processing tasks with a minimum of algorithmic changes and without requiring use of the full original pixel representation.

There is a rich history of multi-resolution and adaptive sampling approaches to image processing, including super-pixels[13,14], wavelet decompositions[15–17], scale-space and pyramid representations[18,19], contrast-invariant level-set representations[20], dictionary-based sparse representations[21], adaptive mesh representations[22–24], and dimensionality reduction[25,26].

However, none of the existing approaches meets all of the above representation criteria, mainly because they were developed for different applications.

Many previous methods, such as super pixels and contrast-invariant level-set representations, provide effective solutions accounting for changes in spatial scales and contrast. They can efficiently be used for specific tasks, such as image segmentation, providing high-quality solutions at reduced memory and computational costs. However, it is unclear how these methods can be used across a wider range of processing tasks, such as image visualization, without still requiring the original pixel image. Alternatively, adaptive sampling methods, such as thresholded wavelets and adaptive mesh methods, provide more general representations that could replace pixel images while reducing both computational cost and memory cost. However, both approaches have not been adapted to account for local contrast variations and are unlikely to be formed rapidly for large 3D images without further improvements. Additionally, techniques that require a change of basis, such as dictionary techniques and wavelets, require the reformulation of image-processing tasks in the transformed domain.

Inspired by the adaptive sampling and local gain control of the human visual system, we here propose a representation of fluorescence microscopy images: the adaptive particle representation (APR). Combining adaptive sampling and local gain control, the APR shares two key features of the human visual system to alleviate current processing and storage bottlenecks in fluorescence microscopy. While the APR reduces storage costs, as data compression also does, it additionally overcomes memory and processing bottlenecks, since the APR can directly be used in processing without going back to pixels. Compression only alleviates storage costs, as the data need to be uncompressed again for processing or visualization. The APR is therefore not a compression scheme, but an efficient image representation that can additionally also be compressed. Here, we present the APR and show that it meets all of the above representation criteria. It, therefore, provides a general framework, combining concepts from the range of existing methods, resulting in an ideal candidate to replace pixel images in fluorescence microscopy.

## Results

**The adaptive particle representation**. The APR adaptively resamples an image, guided by local information content, while using effective local gain control, representing it as a set of particles with associated intensity values. Figure 1a, b illustrates the basic idea of adaptive sampling using a fluorescence image acquired from a specimen of *Danio rerio* with labeled cell nuclei. Particles are a generalization of pixels, i.e., points in space that carry intensity but are not restricted to sit on a regular lattice. Instead, particles can be placed wherever image contents requires, and they may additionally have different sizes in different parts of the image. These sizes define the resolution with which the image is locally represented. The required resolution is given everywhere by an Implied Resolution Function, which attributes high resolution to image areas where the intensity rapidly changes in space (e.g., edges), and low resolution to areas with low variation in intensity (e.g., background or uniform foreground). The Implied Resolution Function defines the radius of a neighborhood around each pixel. Within this neighborhood, the image intensity can be reconstructed at any location by taking a non-negative weighted average of the particles contained in it.

A difficulty in adaptation is to give equal importance to imaged structures across a wide range of intensities. This is achieved by local gain control as illustrated in Fig. 1c–f. Without local gain control, adapting effectively to both bright and dim regions in the

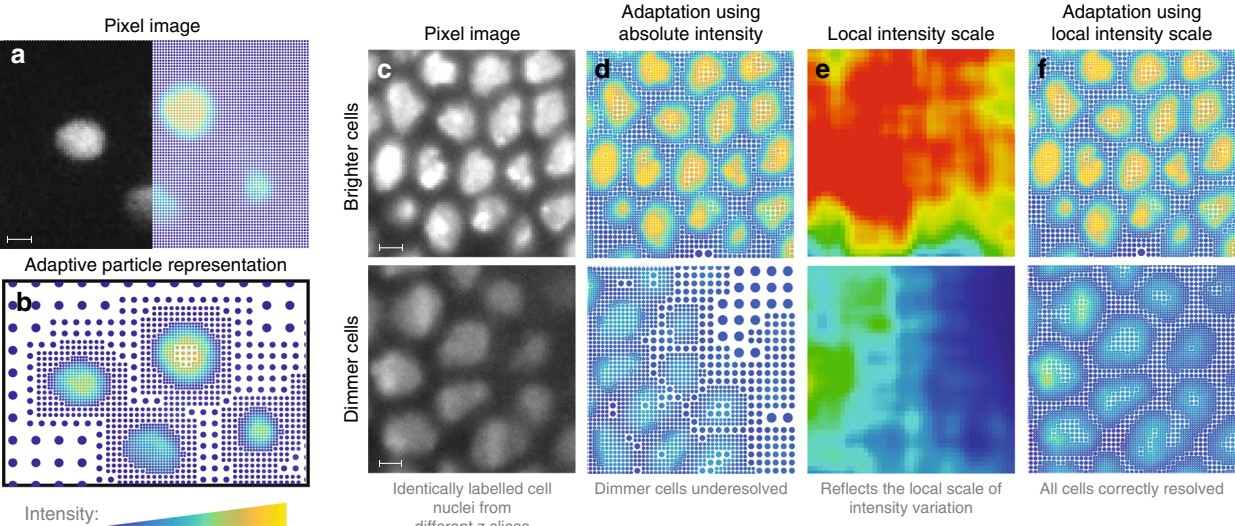

**Fig. 1** Spatially adaptive representation of images. **a** Example image of fluorescently labeled zebrafish cell nuclei (exemplar dataset 7, courtesy of Huisken Lab, MPI-CBG & Morgridge Institute for Research[25]), represented on a regular grid of pixels. **b** The APR of the same image. Particles are shown as dots with their color indicating fluorescence intensity and their size reflecting the local resolution of the representation. **c**–**f** Adaptively representing objects of different intensity requires accounting for the local brightness levels. **c** Two regions of labeled cell nuclei (exemplar dataset 6, courtesy of Tomancak Lab, MPI-CBG) with different brightnesses. **d** Adaptive representation based on the absolute intensity. **f** The APR accounting for the Local Intensity Scale of the image as shown in **e**. In **f**, all objects are correctly resolved across brightness levels. Scale bars indicate 10 pixels

same image is not possible (Fig. 1d). The APR provides local gain control by guiding the adaptation with a local intensity scale (Fig. 1e). As seen in Fig. 1f, this samples dim and bright objects at comparable resolution, giving them equal importance. At the core of this problem is determining the largest possible neighborhood around each pixel such that the reconstructed intensities are guaranteed to be closer to the original intensities than a user-defined threshold $E$, while taking into account a spatially varying Local Intensity Scale $\sigma$. We call this the Reconstruction Condition, requiring that the following inequality holds

$$|I - \hat{I}| \leq E\sigma, \tag{1}$$

for all original pixel locations with intensity $I$ in the original image and intensity $\hat{I}$ reconstructed from the particles within the pixel's local neighborhood.

Here, we present a problem formulation, namely the use of a Resolution Bound and Particle Cells, which together allow us to derive a new algorithm, called the Pulling Scheme, which efficiently finds optimal solutions to the Reconstruction Condition (see Methods). The Pulling Scheme efficiently finds a set of particles, i.e., their locations and intensities, using the magnitude of the intensity gradient of the image and a computed Local Intensity Scale as input, such that the required resolution is guaranteed to satisfy the Reconstruction Condition everywhere. This results in a content-adaptive representation of the image with full user control over the representation quality.

If lossless representation is required, the APR places one particle at each pixel, in which case it becomes equivalent to the original pixel representation. However, fluorescence microscopy images are typically sparse, such that the number of particles can be orders of magnitude less than the number of pixels if small intensity deviations (e.g., within the imaging noise) are allowed. The computational and storage costs of the APR are proportional to the number of particles, and no longer to the number of pixels. By focusing on informative image areas, the APR reduces storage and computational costs and increases the information-to-data ratio.

A didactic introduction to the APR in 1D and details on the formal description, theory, and algorithms can be found in the Methods section and the Supplementary Notes.

**Validation benchmarks**. We validate the APR using noisy synthetic benchmark data in 3D. Supplementary Note 15 and Supplementary Fig. 26 detail the synthetic data generation pipeline. The key advantage of synthetic data is that all relevant image parameters can be controlled and the ground truth image is known. Synthetic images are generated by placing a number of blurred objects into the image domain and corrupting with modulatory Poisson noise. We study the influence of image size, content, and noise level on the performance of the APR. Spherical objects are used for simplicity unless otherwise indicated. Supplementary Note 16 provides a detailed description of each benchmark and the parameters used. All benchmarks use the open-source C++ APR software library LibAPR (available at https://github.com/cheesema/LibAPR) compiled with with gcc 5.4.0 and OpenMP 4.0 shared-memory parallelism on a 10-core Intel Xeon E5-2660 v3 (25 MB cache, 2.60 GHz), 64 GB RAM, running Ubuntu Linux 16.04. Details of the pipeline implementation are given in the Methods section.

In addition to synthetic benchmarks, we also present results for a corpus of 19 exemplar volumetric fluorescence microscopy datasets of different content and imaging modalities, ranging in size from 160 MB to 4 GB. The datasets and parameters used are described in Supplementary Tables 3 and 4 and summary statistics are given in Table 1. Supplementary Figure 33 shows a cross-section of the APR for exemplar dataset 7, and Supplementary Video 1 illustrates the Implied Resolution Function and APR reconstruction for exemplar dataset 1. A comparison of the APR with Haar wavelet thresholding for natural scene images[27] is given in Supplementary Note 12.

We experimentally confirm that the APR satisfies the Reconstruction Condition in Eq. 1 in the absence of noise. Figure 2a shows the empirical relative error $E^* = \left| \frac{I(\mathbf{y}) - \hat{I}(\mathbf{y})}{\sigma(\mathbf{y})} \right|_\infty$ for increasing imposed error bounds $E$, where $\mathbf{y}$ represents all pixel

**Table 1 Performance benchmarks on synthetic and exemplar images**

| | Computational ratio (CR) | Raw image size (GB) | Compressed APR (GB) | MCR of APR | MCR of APR-WNL, $q = 2$ | MCR of pixels-WNL, $q = 2$ | Pipeline time (s) | Pulling Scheme Runtime (s) |
|---|---|---|---|---|---|---|---|---|
| CR5 | 5.63 (0.02) | 1.024 | 0.129 (0.0006) | 7.9 (0.04) | 19.6 (0.97) | 5.4 (0.36) | 2.34 (0.086) | 0.104 (0.002) |
| CR20 | 19.7 (0.13) | 1.024 | 0.036 (0.0002) | 28.4 (0.19) | 64.4 (2.03) | 5.69 (0.62) | 2.01 (0.07) | 0.04 (0.003) |
| CR100 | 93.9 (1.6) | 1.024 | 0.007 (0.0001) | 139.9 (2.1) | 282 (66.4) | 5.85 (0.75) | 1.87 (0.08) | 0.027 (0.005) |
| Exemplars Mean | 51.1 (89.3) | 1.869 (1.38) | 0.051 (0.053) | 129.5 (284) | 297.8 (593) | 95.8 (166) | 3.65 (2.19) | 0.10 (0.08) |
| Exemplars Median | 22.7 | 1.258 | 0.027 | 36.8 | 107.1 | 33.1 | 2.19 | 0.066 |

Results are shown for synthetic images with fixed CR = 5,20,100 and for 19 real-world exemplar datasets (see Supplementary Table 3). For the exemplars, we report the means, standard deviations (brackets), and medians of the values over all exemplar images. For the synthetic fixed-CR benchmarks, the effective CR and the Memory Compression Ratios (MCR) are averaged over image sizes from $200^3$ to $800^3$ (standard deviations in brackets) and the values for absolute runtimes and storage requirements are given for images of size $800^3$. For comparison, we also report the MCR using lossy within-noise-level (WNL) compression[30] of both the APR and the pixel images for the same compression parameter value ($q = 2$, see Supplementary Note 20). We also show the time taken to transform the images to the APR on the benchmark machine and the runtime of the Pulling Scheme alone

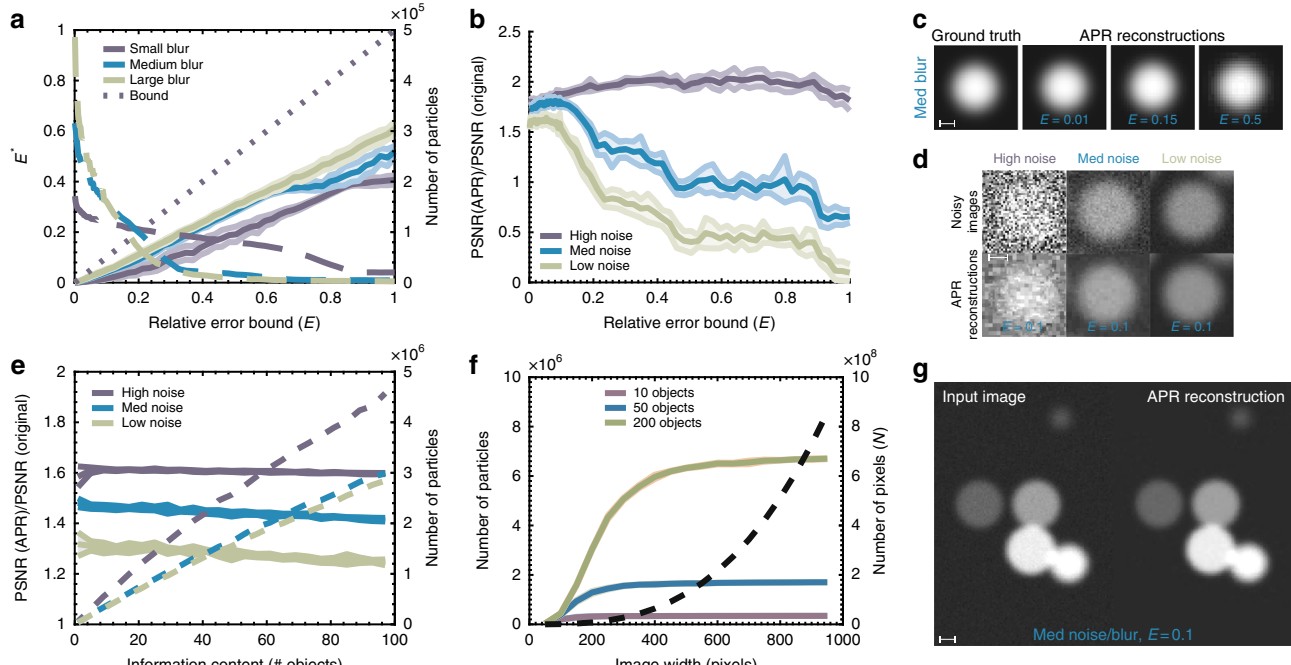

**Fig. 2** Validation of the APR on synthetic data. All results are shown with mean (lines) and standard deviation (bands). **a** Observed reconstruction error $E^*$ (solid lines, left axis) between the ground truth and the piecewise constant APR reconstruction for noise-free images, and number of particles used by the APR (dashed lines, right axis) for different user-defined error bounds $E$ (original image, number of pixels $N = 128^3 \approx 2.1 \times 10^6$). Results are shown for images of different sharpness (blur). The APR reconstruction error is below the specified bound in all cases (dotted line). More accurate APRs require more particles. **b** Peak signal-to-noise ratio (PSNR) of the APR relative to the PSNR of the original pixel image for different error bounds $E$ and image noise levels. For low $E$, the APR has a better PSNR than the input images. **c** APR reconstructions of the medium-blur noise-free test image at different $E$ compared to the ground truth. **d** Examples of test images of spherical objects with different noise levels used in the benchmarks, comparing the original noisy images with their APR reconstructions for $E = 0.1$, illustrating the inherent denoising property of the APR. **e** PSNR ratio (solid lines, left axis) and number of particles used (dashed lines, right axis) for images containing different numbers of objects, i.e., different information content, for $E = 0.1$. In all cases, the PSNR of the APR is better than that of the input image, and the number of particles scales at most linearly with image information content (original image, number of pixels $N = 300^3 = 2.7 \times 10^7$). **f** Number of APR particles (solid lines, left axis) and the fixed number of input image pixels $N$ (black dashed line, right axis) for images of different sizes containing a fixed number of objects ($E = 0.1$). The number of particles plateaus as soon as the objects in the image are well resolved. Scale bars indicate 10 pixels. For additional details, see Supplementary Note 16. **g** Visual comparison of a medium-blur, medium-noise image containing six objects (left) with its piecewise constant APR reconstruction (right) for $E = 0.1$. Note axis units are arbitrary unless otherwise given

locations in the original image and $\sigma(\mathbf{y})$ the Local Intensity Scale (brightness) of the image. In all cases, $E^* < E$, as required by the Reconstruction Condition. As expected, the number of particles used by the APR to represent the image decreases with increasing $E$ (right axis). The results are unchanged when using more complex objects than spheres or different reconstruction methods (Supplementary Fig. 29). Figure 2c provides examples of the

quality of APR reconstruction at different $E$, compared to ground truth. In the absence of noise, the APR satisfies the Reconstruction Condition everywhere, guaranteeing a reconstruction error below the user-specified threshold and fulfilling the first part of RC1.

In real applications, images are corrupted by noise. We find that the introduction of noise introduces a lower limit on the

error $E^*$ that can be achieved (see Supplementary Fig. 30A). This observation agrees with theoretical analysis (Supplementary Note 7). This lower bound is entirely due to the noise in the pixel intensity values, while the adaptation of the Implied Resolution Function is robust to noise. This is demonstrated in Supplementary Fig. 30B, where noisy particle intensities are replaced with ground truth values for the reconstruction step. Adaptation is still done on the noisy pixel data. Then, $E^*$ can be made arbitrarily small, indicating that the construction of the APR is robust against imaging noise. This result also agrees with the theoretical analysis of the impact of errors on the Implied Resolution Function (Supplementary Note 7).

To understand how to best set $E$ in the presence of noise, we compute the observed peak signal-to-noise ratio (PSNR) of the reconstructed image and compare with the PSNR of the original image. Figure 2d provides examples of the different noise levels used. Figure 2b shows that decreasing $E$ to zero does not maximize the PSNR. Instead, for medium to high quality input images, the PSNR is highest between an $E$ of 0.08 and 0.15. For low-quality input images, we find a monotonic relationship between the PSNR and $E$, as de-noising from downsampling dominates. Also, for $E < 0.2$ the reconstruction error is always less than the noise in the input image, reflected in a PSNR ratio greater than one. Therefore, for noisy images with medium to high quality, there is an optimal range for $E$ between 0.08 and 0.15. In this range, the reconstruction errors are less than the imaging noise, and the signal-to-noise ratio of the APR is better than that of the input pixel image, fulfilling also the second part of RC1.

The noise distribution over the particles in the APR depends on the original noise distribution of the pixel image and on the method used to interpolate the particle intensities from the pixels. In Supplementary Note 7, we provide both numerical and theoretical results for the interpolation scheme used here. We consider both Gaussian and Poisson noise on the input pixel image. The variance of the noise scales inversely proportional with the Particle Cell level $l$ (see Methods for definition). For Gaussian noise, the noise remains Gaussian on each level with variance scaled by a factor of $2^{d(l-l_{\max})}$, where $d$ is the image dimension. This is expected, as coarser levels correspond to more averaging and hence noise reduction.

In Fig. 2e we show how the APR adapts to image content. This adaptation is manifested in the linear relationship between the number of objects (spheres) randomly placed in the image and the number of particles used by the APR (right axis). Adaptation is linear despite the brightness of the objects randomly varying over an order of magnitude. Image quality is maintained throughout (left axis). Figure 2g shows an example of a medium-quality input image and its APR reconstruction. Figure 2f shows that the number of particles used by the APR to represent a fixed number of objects becomes independent of image size. Also, if pixel resolution and image size are increased proportionally, the APR approaches a constant number of particles (Supplementary Fig. 31). These results show that the APR adapts proportionally to image content, independent of the number of pixels, fulfilling RC2.

So far, we have not directly assessed the validity of the Local Intensity Scale $\sigma$. In order to do this, we need a ground truth reference. In Supplementary Note 16 we introduce the perfect APR, and the Ideal Local Intensity Scale $\sigma^{\text{ideal}}$ that can be calculated for synthetic data. This ground truth representation is then used to benchmark the APR. The results in Supplementary Tables 1 and 2 show that the Local Intensity Scale we use is effective over a wide range of scenarios. However, for crowded images with large contrast variations (two orders of magnitude or more), we find that the Local Intensity Scale overestimates the

dynamic range of dim regions that are close to bright regions. This effect is most pronounced in high-quality images, where alternative formulations of the Local Intensity Scale could provide better results.

**Performance benchmarks.** Next, we assess the performance of the APR. The APR is adaptive. Therefore, its computational and memory costs depend on image content through the number of particles. We define the Computational Ratio (CR) as:

$$\text{CR} = \frac{\text{number of input pixels}}{\text{number of output particles}}. \tag{2}$$

We assess the performance of the APR for synthetic images with numbers of objects roughly corresponding to $\text{CR} = 5$, 20, 100, representing high, medium, and low complexity images (Supplementary Fig. 32, Supplementary Note 17). For these, the APR achieved effective CR values of 5.63, 19.7, and 93.9, respectively. The results are given in Table 1.

Determining the APR of an image requires approximately 2.7 times (for 16-bit images) the size of the original image in memory. The maximum image size is only limited by available main memory (RAM) of the computer and by the ability to globally index the particles using an unsigned 64-bit integer. Our pipeline has been successfully tested on datasets exceeding 100GB (Supplementary Fig. 35). To test on very large data, exemplar dataset 17 was tiled 200 times to create a 320GB image. Using the same parameters as for the original image resulted in an APR of 4.08GB and a CR of 20.2.

On our benchmark system, we find linear scaling in $N$ and an average data rate of 507MB s$^{-1}$ for transforming images to their APR. This rate corresponds to 3.9 s to form the APR from an input image of size $N = 1000^3$. On the exemplars, execution times range from 0.37 to 8.14 s, with an average of 3.65 s. Table 1 summarizes the results. We find the following distribution of computation time: the Pulling Scheme on average takes <3.5% of the total time, while the computation of the intensity gradient magnitude using smoothing B-splines dominates the execution time, taking up 59% of the total time. For details see Supplementary Note 19.

Our software pipeline shows efficient parallel scaling (Amdahl's Law, parallel fraction = 0.95) on up to 47 cores, achieving data rates of up to 1400MB s$^{-1}$ (Supplementary Fig. 35). This enables real-time conversion of images to the APR, as it is faster than the acquisition rate of current microscopes[28,29].

We conclude that images can be rapidly converted to an APR with a cost that scales at most linearly with image size, fulfilling RC3.

For the fixed-CR datasets, we observe an average Memory Compression Ratio (MCR = (size of the input image file in bytes)/(size of the compressed APR file in bytes)) of 1.4 times the CR. The median MCR of the exemplars is 36.8, and the mean is 129.5. This corresponds to an average size of the input images of 1.87 GB and 51 MB on average for the compressed APR files. Table 1 summarizes the results and Supplementary Table 4 provides the image details.

When the APR is stored as a compressed file, on average 89% of the bytes are used to store the particle intensities, implying that the APR data structures occupy 11% on average. In addition, the APR particle intensities can be compressed further in a lossy manner using existing lossy image compression algorithms. This is shown in Table 1, where we also report the MCR using the within-noise-level (WNL) compression algorithm for large fluorescence images[30] for both the APR and original pixel image. Details on the implementation and benchmarks on synthetic data

are provided in Supplementary Note 20. On synthetic data, we find that the APR and pixel images provide the same image quality after lossy compression, but the APR increases the compression ratio five fold. This indicates that the APR data structures are better suited for further compression using existing compression techniques.

In summary, the APR can be efficiently compressed with a file size proportional to image content, fulfilling RC2. Unlike compression techniques, the APR is an image representation that can be leveraged to accelerate downstream processing tasks, including compression, without reverting to the original pixel image.

**Image processing using the APR**. Image-processing methods are always developed with a certain interpretation of images in mind. Just like pixels, one can also interpret and use the APR in different ways, depending on the processing task. These interpretations align with those commonly used in pixel-based processing. Figure 3a–d outlines the four main interpretations of the APR: collocation points, continuous function approximations, trees, and graphs. Figure 3e–h highlights that while particles store fluorescence intensity, like pixels, they also provide additional information adapted to the image content.

The APR can accelerate existing algorithms in two ways: first, by decreasing the total processing time through reducing the number of operations that have to be executed. Second, by reducing the amount of memory required to run the algorithm. The relative importance of the two, and the degree of reduction, depends on the specific algorithm and its implementation. We quantify the improvements for different algorithms and input images.

We analyze two low-level and one high-level image-processing tasks, namely, neighbor access and filtering as low-level tasks, and image segmentation as a high-level task. The low-level tasks represent a lower bound on the benefits of the APR due to their

simple operations and access patterns, which are best suited for processing on pixels. The segmentation task, in contrast, provides a representative practical example of microscopy image analysis.

For these three benchmarks, we provide results for the computational and memory metrics for three fixed-CR datasets with input images from $N = 200^3$ up to $N = 1000^3$, and for all real-world exemplar datasets. The results of all benchmarks are summarized in Table 2. Supplementary Note 22 describes the benchmark protocols.

The first evaluation metric relates to the computational performance of the algorithm. For a given algorithm and implementation, we define the speed-up (SU) as:

$$SU = \frac{\text{Processing time of the algorithm on pixels}}{\text{Processing time of the algorithm on APR}}. \qquad (3)$$

It is insightful to relate the SU to the CR by SU = CR * (Pixel-Particle Speed Ratio) (PP), where PP = (Time to compute the operation on one pixel)/(Time to compute the operation on one particle). The value of PP depends on many factors, including memory access patterns, data structures, hardware, and the absolute size of the data in memory. Consequently, even for a given algorithm running on defined hardware, the PP is a function of the input image size $N$. For tasks with PP < 1, as in some low-level vision tasks, there is a minimum value of CR above which the algorithm is faster on the APR than on pixels. For tasks with PP > 1, processing on the APR is always faster than on pixels.

For an algorithm on a pixel image, the Memory Cost (MC) in bytes usually scales linearly with the number of pixels $N$ and the number of required temporary and output variables (i.e., copies of the image), as MC = (Number of variables)×(Data type in bytes)×$N$. The memory cost of the APR is: MC = $N_p$×((Number of variables)×(Data type in bytes) + (Cost of APR data structure per particle)), where $N_p$ is the number of particles. We find an

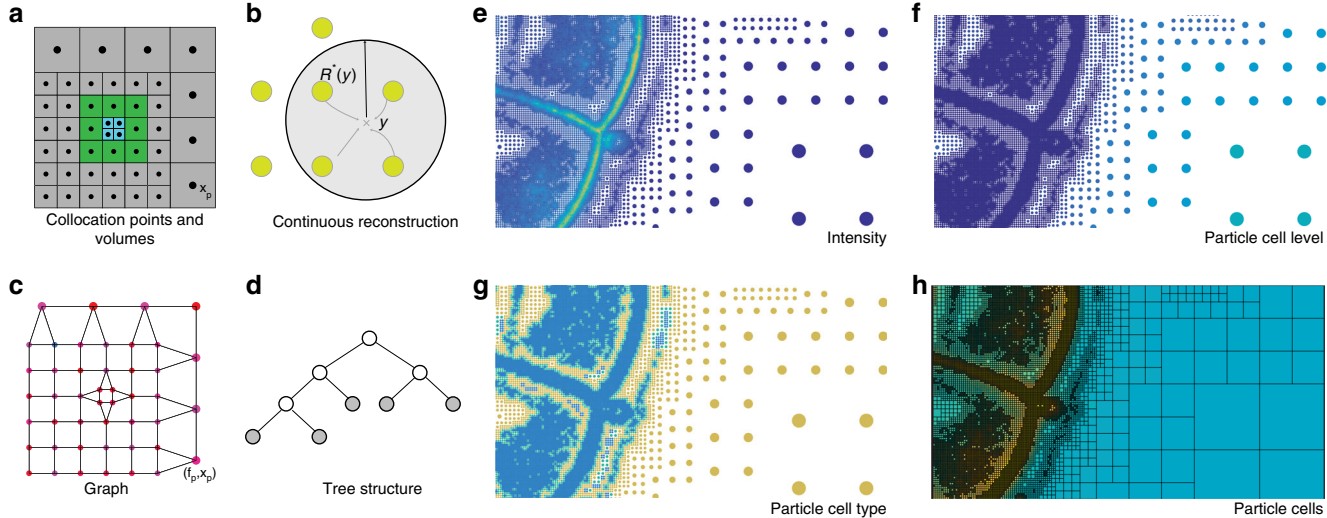

**Fig. 3** Interpretations of the APR for image processing. **a** The APR can be interpreted as a spatial partition defined by the Particle Cells or by the set of particles with positions $x_p$. This interpretation relates to the concept of super-pixels[13]. **b** The APR can be interpreted as a continuous function approximation where the intensity value can be reconstructed at each location $y$, also between particles and pixels, relating to smooth particle function approximations[31]. **c** The APR can be interpreted as a graph, where the particles are vertices and edges link neighboring particles (Supplementary Note 21). This relates the APR to graphical models of pixel images[32]. **d** The APR can be interpreted as a pruned binary tree (quadtree in 2D, octree in 3D) with links between parent and children Particle Cells. This relates the APR to wavelet decompositions[17], image pyramids[19], and tree-based methods[33]. **e–h** While particles store local fluorescence intensity, just like pixels (**e**), they also provide additional information that is not available on the pixels. This includes the Particle Cell level containing information about the local level of detail in the image (**f**), the Particle Cell type encoding the structure of the image (**g**), and the Particle Cell membership providing a content-adaptive disjoint partition of the image domain (**h**). (See Methods section for definitions; original image: exemplar dataset 10[34,35])

**Table 2 Processing benchmarks on synthetic and exemplar images**

|  | Speed up | Time APR (s) | Time pixels (s) | PP | MC pixels (GB) | MC APR (GB) | MRR |
|---|---|---|---|---|---|---|---|
| **Linear neighbor iteration** | | | | | | | |
| CR5 | 0.55 (0.02) | 1.86 (0.07) | 1.02 (0.0002) | 0.097 (0.003) | 3.072 (0) | 0.599 (2.7) | 5.12 (0.02) |
| CR20 | 1.9 (0.09) | 0.54 (0.03) | 1.02 (0.002) | 0.096 (0.004) | 3.072 (0) | 0.181 (1.0) | 16.9 (0.09) |
| CR100 | 7.1 (0.5) | 0.14 (0.009) | 1.02 (0.007) | 0.076 (0.005) | 3.072 (0) | 0.053 (0.0005) | 60.2 (2.3) |
| Exemplars mean | 4.06 (5.7) | 0.86 (0.5) | 1.83 (1.3) | 0.094 (0.01) | 5.61 (4.2) | 0.278 (0.28) | 37.5 (56) |
| **Random neighbor access** | | | | | | | |
| CR5 | 0.71 (0.03) | 15.4 (0.2) | 11.0 (0.4) | 0.126 (0.005) | 3.072 (0) | 0.599 (2.7) | 5.12 (0.02) |
| CR20 | 3.52 (0.3) | 3.23 (0.05) | 11.4 (0.8) | 0.178 (0.01) | 3.072 (0) | 0.181 (1.0) | 16.9 (0.09) |
| CR100 | 24.8 (0.8) | 0.44 (0.01) | 11.01 (0.3) | 0.26 (0.007) | 3.072 (0) | 0.053 (0.0005) | 60.2 (2.3) |
| Exemplars mean | 11.57 (23.6) | 7.29 (10.1) | 21.4 (16) | 0.17 (0.05) | 5.61 (4.2) | 0.278 (0.28) | 37.5 (56) |
| **Image filtering** | | | | | | | |
| CR5 | 7.36 (1.2) | 1.36 (0.009) | 8.02 (0.2) | 1.26 (0.2) | 4.10 (0) | 0.93 (0.003) | 4.38 (0.04) |
| CR20 | 14.82 (3.7) | 0.76 (0.01) | 8.07 (0.3) | 0.77 (0.2) | 4.10 (0) | 0.30 (0.002) | 12.82 (0.9) |
| CR100 | 31.10 (13) | 0.57 (0.003) | 7.96 (0.3) | 0.35 (0.15) | 4.10 (0) | 0.09 (0.0002) | 36.85 (6.7) |
| Exemplars mean | 12.27 (3.0) | 1.24 (0.93) | 14.13 (9.8) | 0.51 (0.33) | 7.48 (5.5) | 0.36 (0.28) | 24.49 (19) |
| **Image segmentation** | | | | | | | |
| CR5 | 5.10 (0.7) | 1.87 (0.02) | 8.86 (0.09) | 0.86 (0.04) | ≈68.5* (0) | 12.57 (0.08) | 5.51 (0.14) |
| CR20 | 18.30 (2.6) | 0.48 (0.003) | 8.83 (0.08) | 0.95 (0.07) | ≈68.5* (0) | 3.75 (0.02) | 18.18 (0.3) |
| CR100 | 85.3 (12) | 0.10 (0.001) | 8.78 (0.09) | 0.97 (0.06) | ≈68.5* (0) | 0.80 (0.003) | 84.09 (2.9) |
| Exemplars mean | N/A | 6.99 (5.9) | N/A | N/A | ≈385* (286) | 13.54 (11.7) | 39.72 (40) |

For the exemplars, we report the means (standard deviation in brackets) of the values over all exemplar images. For the synthetic fixed-CR datasets, the speed-ups (SU), Pixel-Particle Speed Ratios (PP), and Memory Reduction Ratios (MRR) = (Memory Cost Pixels)/(Memory Cost APR) are averaged over image sizes from $200^3$ to $1000^3$; absolute timings and Memory Cost (MC) are given for images of size $800^3$. Graph-cut segmentation on pixels was not possible for $800^3$ images, as the memory requirement exceeded the 64 GB available on the benchmark machine. The corresponding entries in the table (marked with *) are extrapolated from benchmarks run on smaller images and the SU, PP, and pixel timing for the exemplars could not be determined in this case (N/A). See main text and Supplementary Note 22 for a detailed descriptions of the benchmarks

estimated average of 8 bits per particle overhead for the sparse APR data structure. As the number of algorithm variables increases, the overhead of the APR is amortized, so that the reduction in memory cost approaches the CR.

The first task of low-level neighbor access involves averaging the intensities of all face-connected neighbors of a particle or pixel. In the APR, neighbors are defined by the particle graph, as shown in Fig. 3c and described in Supplementary Note 21. We benchmark two forms of neighbor access: linear iteration loops over all neighbors in sequential order; random access visits neighbors in random order, irrespective of how they are stored in memory.

For linear iteration, the APR shows low SU. It is even slower than pixel operations for the synthetic images with CR = 5 and for four of the exemplar datasets (Table 2, group 1). This is because linear iteration is well suited to pixel images. However, the APR provides consistently higher SU for random neighbor access, especially for high CRs. This is likely due to the smaller overall size of the APR improving cache efficiency.

The total memory cost of the APR reflects the CR of the dataset. This provides significant memory cost reductions across all benchmark datasets for both the linear and random neighbor access patterns (see Table 2).

Second, we consider the task of filtering the image with a Gaussian blur kernel. We exploit the separability of the kernel and perform three consecutive filtering steps using 1D filters in each direction. On the APR, this requires locally evaluating the function reconstruction. The benchmark results are shown in Table 2, group 3. Directly filtering the APR consistently outperforms the pixel-based pipeline, both in terms of memory cost and execution time.

In Supplementary Note 22, we analyze the results in detail and find that the APR is most appropriate in cases where the filtered image has a similar structure as the original image, such that the same set of content-adapted particles is suitable to represent both images. Supplementary Figure 40 illustrates this, showing that for a weak blur the APR filter has higher PSNR than the pixel filter. For stronger blurs this is reversed, because the specific APR adapted to the input image is no longer optimal to represent the filtered image.

Finally, we perform binary image segmentation by graph cuts, using the method and implementation of ref. [32]. to compute the optimal foreground/background segmentation for both APR and pixel images. When computing the cut energies, we exploit the additional information provided by the Particle Cell level, type, and local intensity range (see Methods section for definitions). To allow direct comparison with pixel-based segmentation, we interpolate all energies calculated on the APR to pixels and determine the cuts over the pixel image using the same energies. For both APR and pixel images, a face-connected neighborhood graph is used. Given the energy calculations are identical, we benchmark the execution time and memory cost of the graph-cut solver. The results are shown in Table 2, group 4. For the APR we find that the SU directly reflects the CR.

Using the APR, all exemplar images can be segmented without problems, while pixel images can only be segmented for sizes $N \leq 550^3$ on our benchmark machine with 64 GB RAM, illustrating the benefits of the reduced memory cost of the APR.

We validate the APR segmentations by comparing both the APR and pixel-based segmentations to ground truth using the Dice coefficient[38]. Across datasets, we find that the Dice coefficients are not statistically significantly different (p-value: 0.92, Welch's t-test). We provide a representative example in Supplementary Video 2 and show a 3D rendering of a segmentation in Fig. 4e.

The APR provides additional information about the image that is not contained in pixel representations. This information can be exploited in existing image-processing algorithms, as illustrated in the segmentation example above. In addition, it can also be used to design entirely novel, APR-specific algorithms. For example, we define a discrete filter over neighboring particles in the APR particle graph. Since the distances between neighboring particles vary across the image, depending on image content, this amounts to spatially adaptive filtering with the filter size automatically adjusting to the content of the image. On the APR, this only requires linear neighbor iteration, while an adaptive pixel implementation would be significantly more complex.

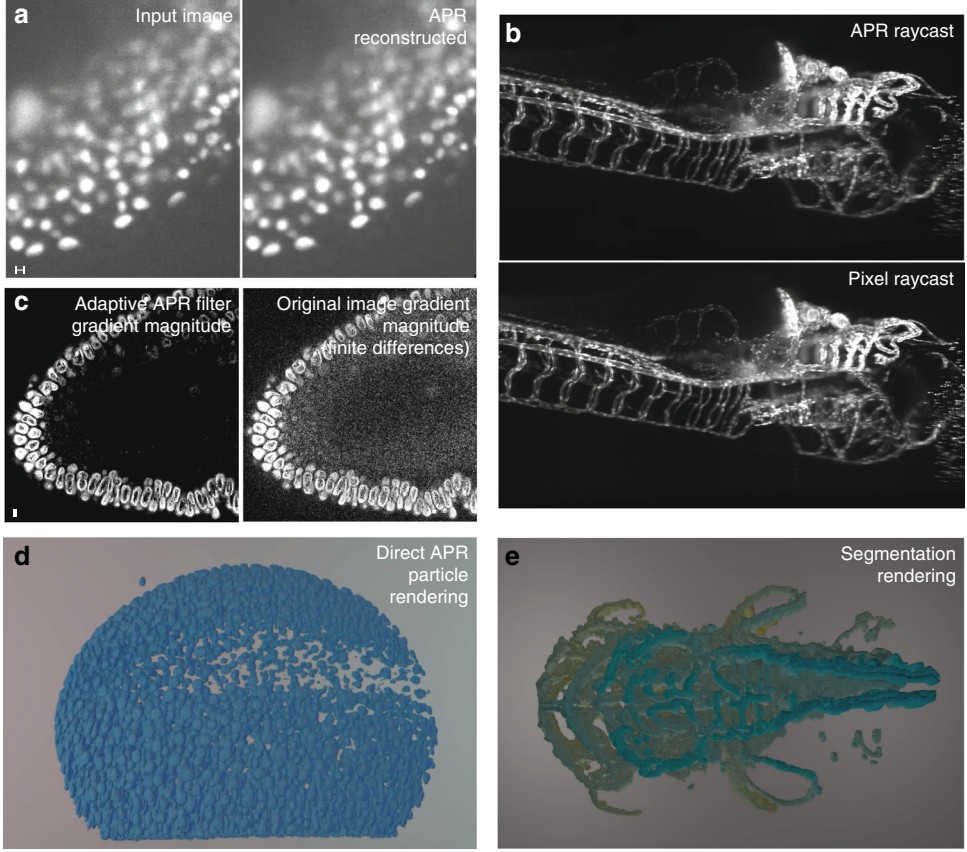

**Fig. 4** Image processing using the APR. **a** Comparison of an example image (exemplar dataset 7[25]) with its piecewise constant APR reconstruction. **b** Comparison of the maximum-intensity projection of a direct 3D APR raycast with the maximum-intensity projection of the pixels for exemplar dataset 17 (full image see Supplementary Fig. 45). **c** Comparison of the intensity-gradient magnitude estimated using the Adaptive APR Filter and using central finite differences over the pixels for exemplar dataset 6 (Tomancak Lab, MPI-CBG). **d** Direct 3D particle rendering of zebrafish nuclei (exemplar dataset 7[37]) using Scenery. (Raw images in **a**, **b**, **d**, **e** courtesy of Huisken Lab, MPI-CBG & Morgridge Institute for Research.) Scale bars indicate 10 pixels. Additional details can be found in Supplementary Note 22. **e** APR volume rendering of a 3D image-segmentation result, colored by depth using the open-source visualization tool Scenery[36], computed using graph-cut segmentation directly on the APR, as described in Supplementary Note 22 (exemplar dataset 13[37]). Segmentation on the APR took 5.5 s. Segmentation on the pixel image was impossible on our benchmark machine due to memory overflow

Supplementary Figure 42 shows synthetic results for an adaptive blurring filter, and Supplementary Fig. 43 for a filter that adaptively estimates the intensity gradient magnitude. In both examples, the adaptive APR-filtered results have higher PSNR than results from corresponding non-adaptive pixel filters, as shown in Fig. 4c.

Images represented using the APR can directly be visualized without going back to pixels. APR images can be visualized using both traditional and novel visualization methods. We provide examples of three visualization methods. Figure 4a and Supplementary Video 1 show examples of a slice-wise APR reconstruction in comparison with the pixel images. Figure 4b and Supplementary Video 3 show a perspective APR maximum-intensity projection in comparison with the same ray-cast of the original pixel image. The resulting visualizations are visually indistinguishable. APR raycasting only requires storing and computing on the APR, reducing memory and computational costs proportionally to the CR of the image, enabling full-scale visualization of large images. Lastly, we can directly visualize the particles of an APR as glyphs both in 2D, see Figs. 1, 6, and 3 and in 3D, see Fig. 4e and d and Supplementary Videos 4 and 5 (images from refs 39 and 25, respectively, and rendered using the open-source 3D visualization software Scenery[36]).

In summary, the APR reduces memory and computational costs of downstream processing tasks without requiring going back to the full pixel image, fulfilling RC4.

## Discussion

We have introduced a content-adaptive image representation for fluorescence microscopy, the APR. The APR is inspired by how the human visual system effectively avoids the data and processing bottlenecks that plague modern fluorescence microscopy, particularly for 3D imaging. The APR combines aspects of previous adaptive-resolution methods, including wavelets, super-pixels, and equi-distribution principles in a way that fulfills all Representation Criteria (RC) set out in the introduction. The APR is computationally efficient, suited for real-time applications at acquisition speed, and easy to implement.

We presented the ideas and concepts of the APR. The APR resamples an image by adapting a set of particles to the content of the image, taking into account the Local Intensity Scale, similar to gain control in the human visual system. The main theoretical and algorithmic contribution that made this possible with a computational cost that scales linearly with image size is the Pulling Scheme. The Pulling Scheme guarantees image representations within user-specified relative intensity deviations.

We verified accuracy and performance of the APR using synthetic benchmark images. The analysis showed that all theoretical results hold in practice, and that the number of particles used by the APR scales with image content, while maintaining image quality (RC1). Further, we showed that although image noise

places a limit on representation accuracy, there exists an optimal range for the relative error threshold. In this range, the reconstruction error for noisy images is always well within the imaging noise level (RC1). Moreover, we found that the number of particles is independent of the original image size, with computational and memory costs of the APR instead proportional to the information content of the image (RC2). We showed how pixel images can rapidly be transformed to the APR, and efficiently stored both in memory and in files (RC3). We have demonstrated that the APR benefits, both in terms of execution time and memory requirements, can be leveraged for a range of image-processing tasks without ever returning to a pixel image, with minimal changes to existing algorithms (RC4). Across all benchmarks and exemplar datasets, other than the worst-case example of linear neighbor access, processing directly on the APR resulted in lower execution times and memory costs. Moreover, in the examples of visualization and segmentation, the memory cost reduction of the APR enabled processing of data sets that would not otherwise have been possible on our benchmark machine.

The APR has a range of interpretations that align with those of pixel images, allowing direct application of established image-analysis frameworks to the APR. For algorithms that require a locally isotropic neighborhood, the anisotropic local neighborhood of the APR graph can be hidden by using a particle-wise isotropic patch reconstruction, enabling also these algorithms to directly run on the APR with minimal changes. In many cases, however, the additional information about the structure of the image provided by the APR can improve existing algorithms, as shown here for segmentation, and it can be used to design novel algorithms, such as content-adaptive filters (shown here), adaptive variational models[40], and Laplacian graph-based image processing methods[41].

When designing APR-based algorithms, it may be important to account for the noise distribution[42]. As expected, the noise distribution on the particles is different from that of the original pixels, transformed by the interpolation scheme used to compute particle intensities, and it naturally decomposes by resolution level (see Supplementary Note 7). Therefore, noise terms or regularizers in image-processing models may have to be adjusted or designed accordingly. However, the noise distribution in content-rich areas, notably around edges in the images, is largely unchanged. For image-analysis methods that focus on these areas, such as segmentation methods, the same noise models as on pixels may thus still be used.

Taken together, the APR meets all four representation criteria set out in the Introduction. We believe that the gains of the APR will in many cases suffice to relax the current processing bottlenecks. In particular, image-processing pipelines using the APR would be well suited for high-throughput experiments and real-time processing, e.g., in smart microscopes[9,39]. However, the APR is sub-optimal with respect to the number of particles used. This sub-optimality results from the conservative restrictions required to derive the efficient Pulling Scheme, and from the generality of the Reconstruction Condition. It is proven by the fact that the APR particle properties can be represented by a Haar wavelet transform[17] with a number of non-zero coefficients that is either equal to or less than the number of particles in the APR, while allowing exact reconstruction of the APR particle properties (Supplementary Note 12).

The use of adaptive representations of images[22–24] and its motivation by the human visual system[13,18] are not new. While the adaptive placement of the particles in the APR bears visual similarity to half-toning methods and techniques based on the Floyd-Steinberg error-diffusion algorithm[43], the mathematical foundations and the algorithms themselves differ fundamentally.

The APR does, however, share several concepts with established adaptive representations. The Resolution Function $R(y)$ of the APR, e.g., is related to the oracle adaptive regression method[44], and the derivation and form of the Resolution Bound are related to ideas originally introduced in equi-distribution methods for splines[45–47], which also inspired the work here[48]. The Reconstruction Condition for a constant Local Intensity Scale relates to infinity-norm adaptation for wavelet thresholding as used in adaptive surface representations[49]. Finally, powers-of-two decomposition of the domain is central to many adaptive-resolution methods[17,19,33,50] and its use here was particularly inspired by ref. [51]. Despite these relations to existing methods, the APR uniquely fulfills all representation criteria and extends or links many of the previous concepts.

Core novelties of the APR include the spatially varying Local Intensity Scale, the broad class of reconstruction methods available, "backwards compatibility" to pixel images by on-the-fly local patch reconstruction, guaranteed theoretical bounds on the representation accuracy, and the ability to combine existing compression schemes with the APR. Moreover, the computational efficiency of the APR enables real-time workflows where images are transformed at acquisition rate.

The APR has the potential to replace pixel-based image-processing pipelines for the next generation of fluorescence microscopes. We envision that the APR is immediately formed, possibly after image enhancement[52], on the acquisition computer or even on the camera itself. Following this, all data transfer, storage, visualization, and processing can be done on the APR, relaxing downstream bottlenecks. In cases where regulatory requirements or statistical noise analyses require the raw pixel data to be archived, this is best done by archiving the difference image between the raw pixel data and the APR. Since the APR captures all imaged structures, the difference image is typically very sparse and can effectively be compressed using lossless methods. All processing can then still be done on the APR from which the raw pixel data can exactly be reconstructed using the archived difference image whenever needed.

The full realization of APR-based pipelines requires further algorithm and software development, including GPU acceleration, block-wise APR transforms for images that exceed available computer memory, and integration with current microscope systems, image databases[53], and image-processing tools[54]. This integration is enabled through wrappers of the provided C++ Library LibAPR (see Methods section).

Here, we presented a particular realization of an APR pipeline. We foresee alternative pipelines, e.g., using deep learning approaches[55] to improve estimation of the Local Intensity Scale, of the image intensity gradient, and smooth image reconstructions. Just as in space, the APR can also be used to adaptively sample time. Such temporal adaptation can lead to a multiplicative reduction in memory and computational costs compared to those presented here. Further, the APR can be extended to allow for anisotropic adaptation using rectangular particle cells, local affine transformations, and anisotropic particle distributions within each cell.

Given the wide success of adaptive representations in scientific computing, the unique features of the APR could be useful also in non-imaging applications. This includes applications to time-series data, where the APR could provide an adaptive regression method[44], to surface representation in computer graphics[49], and to numerically solving partial differential equations with spatial adaptivity[48,56–58].

## Methods

**APR theory and algorithms**. We describe the basic concepts of the APR and its components. We provide all technical details needed to reproduce or reimplement

the APR. For simplicity, we do so using a 1D image as a didactic example (see also Supplementary Note 11; code available from https://github.com/cheesema/APR_1D_demo). All concepts extend to higher dimensions and to higher derivatives, as shown in Supplementary Notes 1, 3, and 9.

For the APR to optimally represent a given image with intensities $I(y)$ at pixels $y$, the Implied Resolution Function should be as large as possible at every location, while still guaranteeing that the image can be reconstructed within the user-specified relative error $E$ scaled by the Local Intensity Scale $\sigma(y)$. The Local Intensity Scale $\sigma(y)$ is an estimate of the range of intensities present locally in the image. Considering an arbitrary Resolution Function $R(y)$, we can formulate the problem as finding the largest $R(y)$ everywhere that satisfies

$$|I(y) - \hat{I}(y)| \leq E\sigma(y), \qquad (4)$$

for each pixel $y$, where $\hat{I}(y)$ is the reconstructed intensity calculated by any non-negative weighted average over particles within $R(y)$ distance of $y$. We call this the Reconstruction Condition. For the 1D example shown in Figure 5a, b, a constant local intensity scale $\sigma(y) = 1$ is used. Maximizing $R(y)$ minimizes $\frac{1}{R(y)}$, which is proportional to the locally required sampling density. Therefore, maximizing $R(y)$ results in the minimum number of particles used. Unfortunately, finding the optimal $R(y)$ that satisfies the Reconstruction Condition for arbitrary images requires a number of compute operations that is proportional to the square of the number of pixels $N$. This computational cost is prohibitive even for modestly sized images. We propose two conservative restrictions on the problem and show that the optimal solution to the restricted problem can be computed with a total number of operations that is proportional to $N$.

We outline the two problem restrictions, and how they are used to formulate an efficient linear-time algorithm for creating the APR.

The first restriction on the Resolution Function $R(y)$ requires that for all original pixel locations $y$, it satisfies the inequality

$$R(y) \leq L(y^*) \qquad (5)$$

for all $y^*$ with $|y - y^*| \leq R(y)$, and $L(y) = \frac{E\sigma(y)}{|\nabla I|}$. Here $|\nabla I|$ is the magnitude of the image intensity gradient, which in 1D is $\left|\frac{dI}{dy}\right|$ and can be computed directly from the image. We call this inequality the Resolution Bound, and $L(y)$ the Local Resolution Estimate. If we assume the continuous intensity distribution underlying the image to be differentiable everywhere and the Local Intensity Scale $\sigma(y)$ to be sufficiently smooth (see Supplementary Notes 2 and 3), satisfying the Resolution Bound guarantees satisfying the Reconstruction Condition. In Fig. 5c, we illustrate that the Resolution Bound in 1D requires that a box centered at $y$ with height $R(y)$ and width $2R(y)$ does not intersect anywhere with the graph of $L(y)$. Since the Resolution Bound represents a tighter bound than the Reconstruction Condition, the optimal solution to the Resolution Bound $R_b(y)$ is always less than or equal to the optimal solution to the Reconstruction Condition $R_c(y)$, therefore providing the same or a higher image representation accuracy. The dashed lines in Fig. 5d illustrate this for the 1D example. As mentioned above, solving for the optimal Resolution Function requires computer time $\propto N^2$. However, we show next that the Resolution Bound can be satisfied optimally with computer time linear in $N$, if we add a second restriction.

The second restriction is that the blocks constituting the Resolution Function must have edge lengths that are powers of 1/2 of the image edge length. The piecewise constant Resolution Function defined by the uppermost edges of these blocks is called the Implied Resolution Function $R^*(y)$ and is shown in black in Fig. 5d. The blocks we call Particle Cells. They have sides of length $\frac{|\Omega|}{2^l}$, where $|\Omega|$ is the edge length of the image, measured in pixels. The number $l$ is a positive integer we call the Particle Cell Level. Each Particle Cell $c_{i,l}$ is therefore uniquely determined by its level $l$ and location $i$. Figure 5d inset illustrates these definitions for a single Particle Cell (see Supplementary Note 4 for the nD formal definition). The size of the blocks on the lowest resolution level is half the size of the image ($l_{min} = 1$), and the highest level of resolution $l_{max}$ contains boxes the size of the original pixels. For image edge lengths that are not powers of 2, the parameter $|\Omega|$ is rounded upwards to the nearest power of two without padding the image.

Thanks to these two restrictions, the problem of finding the optimal Resolution Function can be reduced to finding the smallest set $\mathcal{V}$ of Particle Cells that defines an Implied Resolution Function $R^*(y)$ that satisfies the Resolution Bound (Supplementary Note 4). We call this minimal set $\mathcal{V}$ of Particle Cells the Optimal Valid Particle Cell (OVPC) set. In Supplementary Note 8, we provide additional analysis of the impact of these restrictions on the efficiency of adaptation.

In order to construct an algorithm that efficiently finds the OVPC set for a given Local Resolution Estimate $L(y)$, we first formulate the Resolution Bound in terms of Particle Cells. This formulation requires arranging the Particle Cells $c_{i,l}$ by level $l$ and location $i$ in a tree structure, as shown in Fig. 5e. In 1D this is a binary tree, in 2D a quadtree, and in 3D an octree. When arranged as a tree structure, we can naturally define children and neighbor relationships between Particle Cells, as shown in green and blue, respectively, in Fig. 5e. We also define the descendants of a Particle Cell as the set of all children and children's children up to the maximum resolution level $l_{max}$. Given these definitions, the Local Resolution Estimate $L(y)$ can be represented as a set of Particle Cells $\mathcal{L}$ by iterating over all pixels $y$, and adding the Particle Cell with level $l = \left\lceil \log_2 \frac{|\Omega|}{L(y)} \right\rceil$ and location $i = \left\lfloor \frac{2^l y}{|\Omega|} \right\rfloor$ to $\mathcal{L}$ if it is not

already in $\mathcal{L}$ (assuming the lower-left boundary of the image is at zero). Figure 5f illustrates how $\mathcal{L}$ relates to $L(y)$, with $\mathcal{L}$ also represented in Fig. 5e in the tree structure. We call this set of Particle Cells the Local Particle Cell (LPC) set $\mathcal{L}$ (see Supplementary Note 4).

We can then represent the Resolution Bound in terms of $\mathcal{L}$. A set of Particle Cells $\mathcal{V}$ will define an Implied Resolution Function that satisfies the Resolution Bound for $L(y)$, if and only if the following statement is true: for every Particle Cell in $\mathcal{V}$, none of its descendants, or neighbors' descendants, are in the LPC set $\mathcal{L}$ (Theorem 1 in Supplementary Note 4). We call any set of Particle Cells satisfying this statement "valid". The OVPC set $\mathcal{V}$ is then uniquely defined as the valid set for which replacing any (combination of) Particle Cells with larger Particle Cells would result in $\mathcal{V}$ no longer being valid (Theorem 2 in Supplementary Note 4).

We present an efficient algorithm for finding the OVPC set $\mathcal{V}$, called the Pulling Scheme. The name is motivated by how a single Particle Cell in $\mathcal{L}$ pulls the resolution function down to enforce smaller Particle Cells across the image. The Pulling Scheme finds the OVPC set $\mathcal{V}$ directly, without explicitly checking for validity or optimality. The result is by construction guaranteed to be valid and optimal. In order to derive the algorithm, we leverage three properties of OVPC sets:

1. Predictable and self-similar structure: Neighboring Particle Cells never differ by more than one level and are arranged in a fixed pattern around the smallest Particle Cells in the set. This local structure is independent of absolute level $l$ and endows the set with a self-similar structure. Using this structural feature, the OVPC set $\mathcal{V}$ for a LPC set $\mathcal{L}$ with only one Particle Cell $c_{i,l}$ can be generated directly for any $i$ and $l$ (see Supplementary Fig 3).
2. Separability: We can find the OVPC set given a LPC set $\mathcal{L}$ by considering each cell in $\mathcal{L}$ separately and then combining the smallest Particle Cells from all sets that cover the image (see Lemma 1 in Supplementary Note 5). Supplementary Figure 4 illustrates this separability property.
3. Redundancy: The redundancy property tells us that when constructing $\mathcal{V}$, we can ignore all Particle Cells in $\mathcal{L}$ that have descendants in $\mathcal{L}$. This is because descendants provide equal or tighter constraints on the resolution function than their parent Particle Cells (see Lemma 2 in Supplementary Note 5 for the proof).

These properties enable us to efficiently construct $\mathcal{V}$ by propagating solutions from individual Particle Cells in $\mathcal{L}$, one level at a time, starting from the highest resolution level ($l_{max}$) of the smallest Particle Cells in $\mathcal{L}$. Here, we use a simple implementation that explicitly represents all possible Particle Cells in an image pyramid structure. Alternative implementations are possible that do not require the explicit storage of the full tree structure, but are not discussed here. The Pulling Scheme is summarized in Algorithm 1 in Supplementary Note 5, and Supplementary Fig. 7 illustrates the steps for each level. Supplementary Notes 5 and 13 provide additional details. The computational cost of the algorithm scales with the number of Particle Cells in $\mathcal{V}$. Further, computing the OVPC set $\mathcal{V}$ using the Pulling Scheme incurs a computational cost that is proportional to the number of pixels $N$ for a fixed information-to-data ratio. A comparison of the computational cost of the Pulling Scheme with a greedy approach is given in Supplementary Fig. 12 and Supplementary Note 8.

Using the Equivalence Optimization (see Supplementary Note 5), the computational and memory costs of the Pulling Scheme can be further reduced by a factor of $2^d$, where $d$ is the image dimensionality, while obtaining the same solution. A second optimization restricts the neighborhood of particle cells to further reduce the total number of particles used, as described in Supplementary Note 5. We use both optimizations for the results presented in this paper. Ultimately, the only operations that need to be computed on the full pixel image are the simple filters for the gradient magnitude and the Local Intensity Scale.

Given the Implied Resolution Function computed by the Pulling Scheme, the last step of forming the APR is to determine the locations of the particles $\mathcal{P}$. Locations must be chosen so that around each pixel $y$ there is at least one particle within a distance of $R^*(y)$. This requirement is easily satisfied by placing one particle at the center of each Particle Cell in $\mathcal{V}$. Specifically, for each Particle Cell $c_{i,l}$ in $\mathcal{V}$, we add a particle $p$ to $\mathcal{P}$ with location $y_p = \frac{|\Omega|}{2^l}(i + 0.5)$. For each particle $p$ we store the image intensity at that location $I_p = I(y_p)$, interpolated from the original pixels as described in Supplementary Note 6. This way of arranging the particles has the advantage that the particle positions do not need to be explicitly stored, as they are determined by $\mathcal{V}$.

In Fig. 6, we summarize the steps required to form the APR from an input image. The APR can be stored as the combination $\{\mathcal{V}, \mathcal{P}\}$. We represent the OVPC set $\mathcal{V}$ by storing the integer level $l$ and the integer location $i$ for each Particle Cell. $\mathcal{V}$, therefore, defines the Implied Resolution Function $R^*(y)$ for all $y$ in the image. $\mathcal{P}$ stores the intensities of all particles.

Determining $L(y)$ requires computing the intensity gradient $\nabla I$ over the input image. In practice, the pixel intensities are noisy, which leads to uncertainty in the computed $L(y)$. In Supplementary Note 7, we provide theoretical results on how this uncertainty imposes a lower bound on the achievable representation accuracy $E$.

**3D fluorescence APR implementation**. When implementing the APR, three design choices have to be made: First, one has to decide how to calculate the

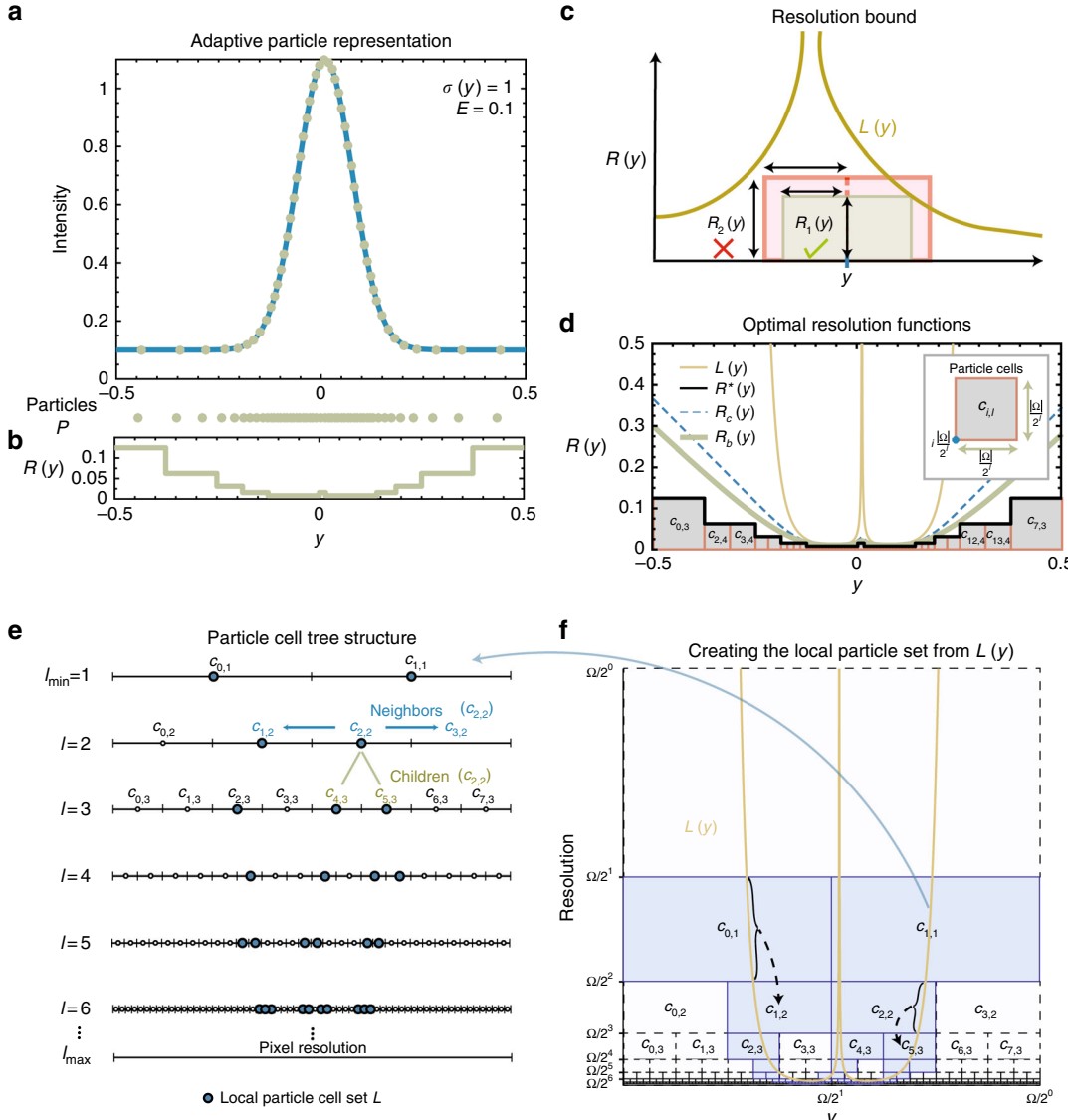

**Fig. 5** Concepts and definitions of the APR illustrated in 1D. **a** APR ($E = 0.1$, $\sigma(y) = 1$) representation of the shifted 1D Gaussian $I(y) = exp\left[\frac{-(y-0.01)^2}{0.009}\right] + 0.1$. Dots indicate particles and the blue line a linear interpolation. **b** The corresponding Resolution Function $R(y)$ with the set of particles $\mathcal{P}$ shown as dots above. Note: all resolution functions are in the same arbitrary spatial unit as $y$. **c** Illustration of the Resolution Bound, requiring for all pixel locations $y$ that a rectangle centered at $y$ of width $2R(y)$ and height $R(y)$ does not intersect the graph of the Local Resolution Estimate $L(y)$. Fulfilling the Resolution Bound guarantees fulfilling the Reconstruction Condition. **d** Comparison of the optimal (i.e., largest everywhere still satisfying the Reconstruction Condition) Resolution Function $R_c(y)$ (dashed blue) with the $R_b(y)$ satisfying additionally also the Resolution Bound (bold green) and with the Implied Resolution Function $R^*(y)$ (bold black) for the 1D Gaussian example from **a**. The Implied Resolution Function is composed of the upper edges of blocks called Particle Cells (gray). They never intersect the optimal Resolution Function ($R_c$), therefore providing a conservative approximation. The inset shows how Particle Cells are described by their level $l$ and location $i$. **e** The set of all possible Particle Cells can be represented as a binary tree reaching down to single-pixel resolution. **f** The Local Particle Cell set $\mathcal{L}$ is constructed from $L(y)$. The correspondences between segments of $L(y)$ and the Particle Cells in $\mathcal{L}$ are shown with braces and dashed lines. All possible Particle Cells are shown as blocks, and those belonging to $\mathcal{L}$ are shaded blue ($\Omega = |\Omega|$ in axes labels for brevity)

intensity gradient magnitude $|\nabla I(y)|$. Second, one has to decide how to compute the Local Intensity Scale $\sigma(y)$. Third, one has to decide how to interpolate the pixel intensities to particle locations to determine particle intensities $I_p = I(\mathbf{y}_p)$. Full details are given in Supplementary Note 13.

To calculate the intensity gradient magnitude over the input image, we use smoothing cubic B-Splines[59], which provide robust gradient estimation in the presence of noise. They require setting a smoothing parameter $\lambda$ depending on the noise level, as described in Supplementary Note 13.

For the Local Intensity Scale $\sigma(\mathbf{y})$, we use a smooth estimate of the local dynamic range of the image, as described in Supplementary Note 13. This form of the Local Intensity Scale accounts for variations in the intensities of labeled objects, similar to gain control in the human visual system. We ensure that $\sigma$ is sufficiently smooth (see Supplementary Note 2) by computing it over the image downsampled by a factor of two. Examples are shown in Figs. 1e and 6c. The size of the smoothing window is given by a rough estimate of the half width at half maximum of the point-

spread function (PSF) of the microscope. Further, a lower threshold is introduced to prevent resolving background noise (see Supplementary Note 13).

Two methods are combined to interpolate pixel intensities to particle locations: for particles in Particle Cells at pixel resolution, the intensities are directly copied from the respective pixels, while for particles in larger particle cells, we assign the average intensity of all pixels in that Particle Cell[19].

We also provide a method for reconstructing a pixel image $\hat{I}(\mathbf{y})$ from the APR. A pixel image satisfying the Reconstruction Condition in Eq. 1 can be reconstructed from the APR using any non-negative weighted average of particles within $R^*(y)$ of pixel $y$. In Supplementary Note 10, we discuss possible weight choices, providing examples of smooth, piecewise constant, and worst-case reconstructions. For displaying figures and benchmarking, unless otherwise stated, we use the piecewise constant reconstruction in this paper. This reconstruction sets all pixels inside a Particle Cell equal to the intensity of the particle in that cell and thus has the best computational efficiency.

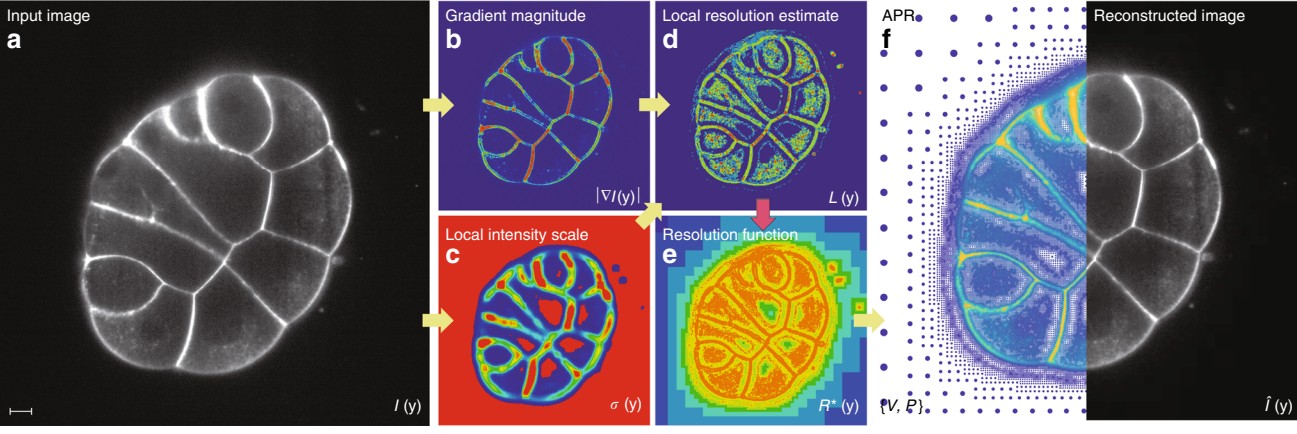

**Fig. 6** Procedure for forming the APR in 3D. **a** 2D slice of exemplar dataset 10 (courtesy of Lemaire lab, CRBM (CNRS) and Hufnagel lab, EMBL[34,35]). **b** First, the intensity gradient magnitude $|\nabla I(\mathbf{y})|$ and **c** the Local Intensity Scale $\sigma(\mathbf{y})$ are calculated from the input image. These two are then combined to compute the Local Resolution Estimate $L(\mathbf{y})$ (**d**). **e** The Pulling Scheme (red arrow) uses $L(\mathbf{y})$ to compute the optimal Implied Resolution Function $R^*(\mathbf{y})$. **f** This is then used to define the Optimal Valid Particle Cell set $\mathcal{V}$ and the particle locations $\mathcal{P}$, which together form the APR. The APR is visualized both as particles with color encoding intensity and size encoding local resolution, and as a piecewise constant reconstruction $\hat{I}(\mathbf{y})$ of the image. Scale bar indicates 40 pixels

All design decisions have been made to optimize robustness against imaging noise and computational efficiency. We find that the method is stable with respect to the choice of parameters. A discussion of parameter selection for real datasets is given in Supplementary Note 14, and the parameter values used for our test datasets are given in Supplementary Table 3.

Appropriate data structures must be used to store and process on the APR efficiently. Ideally, these structures allow direct memory access at low overhead. Here, we propose a multi-level data structure for the APR, as described in Supplementary Note 18. Each APR level $l$ is encoded similar to sparse matrix schemes with Particle Cell locations $\{i_x, i_y, i_z\}$. This data structure efficiently stores $\mathcal{V}$ and $\mathcal{P}$ by explicitly encoding only one spatial coordinate ($i_y$) per Particle Cell, while allowing random access. We call this data structure the Sparse APR (SA) data structure. It relies on storing one red-black tree of Particle Cell locations $i_y$ for each combination of $\{i_x, i_z, l\}$, caching access information for contiguous blocks of Particle Cells. When storing image intensity using 16 bits, the SA data structure requires approximately 50% more memory than the uncompressed particle intensities alone. Simpler data structures, without the red-black tree, can be used to reduce this overhead if random access is not required. In all results presented here, we use the SA data structure.

We store the APR SA data structure using the HDF5 file format[60] and the BLOSC HDF5 plugin[61] for lossless Zstd compression of the Particle Cell and intensity data in the file.

**Code availability**. Code is available through the open-source C++ APR software library LibAPR[62] (available at https://github.com/cheesema/LibAPR), including basic Python wrappers, and Java wrappers can be found at https://github.com/krzysg/LibAPR-java-wrapper. Didactic MATLAB code for the APR in 1D can be found at https://github.com/cheesema/APR_1D_demo.

## Data availability
The data that support the findings of this study are available from the corresponding author upon reasonable request.

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

## Acknowledgements

We thank the members and leaders of the Tomancak Lab at the Max Planck Institute of Molecular Cell Biology and Genetics (MPI-CBG), Huisken Lab at the MPI-CBG and Morgridge Institute for Research, Royer Lab at the CZ Biohub, Keller Lab at HHMI Janelia Farm, Lemaire Lab at Centre des Recherches de Biochimie Macromoléculaire, and the Deutsches Zentrum für Neurodegenerative Erkrankungen e.V., all for generously allowing us to use their images during the development and benchmarking of this work. Further, we thank Joel Jonsson for work on the Python wrappers, Michael Hecht for discussions regarding mathematical notation, and Jan Huisken for his feedback during the development of the APR. This work was funded by the Max Planck Society and by the German Federal Ministry of Education and Research (BMBF) under funding code 031L0044. B.L.C. further acknowledges financial support though a DIGS-BB fellowship, awarded by the DFG-funded Excellence Graduate School of TU Dresden under code DFG-GSC-97.

## Author contributions

The project was conceived by I.F.S. and B.L.C., APR theory and algorithms developed by B.L.C., software and implementation by U.G., M.S., K.G., and B.L.C., benchmarking by B.L.C. and K.G., and the manuscript was written by I.F.S., U.G., K.G., and B.L.C.

## Additional information

**Competing interests:** The authors declare no competing interests.

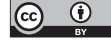

