## [Peer Review File · Nature Communications]

Reviewers' comments:

Reviewer #1 (Remarks to the Author):

SUMMARY:

The paper presents a novel content-adaptive representation ("Adaptive Particle Representation – APR) for 2D-3D fluorescence microscopy image manipulation, visualization, and processing. The potential of APR is demonstrated with theory and experiments on real image data. The paper is well written, and the results are interesting. The properties of APR are somewhat too technical in the first part of the paper. This part is only accessible to specialists in image analysis and image representation.

COMMENTS:

1. The state of the art is partially covered and the manuscript ignores prior relevant work on similar representations. The authors focus on super-pixels and wavelet-based image representation but do not explicitly mention recent work about sparsity-based image representation: many microscopy images are composed of fluorescent objects against a dark background as illustrated in the paper and videos. Another category of approaches in image representation focused on image level lines according to the Matheron's theory. The Fast Level Set Transform (Monasse and Guichard, 2000) is a typical algorithm used to decompose the image into a tree of shapes with no loss of information. Finally, I found that the APR concept can be seen as generalization of the popular halftoning (and dithering) method introduced several decades ago for image printing. The basic idea is to simulate shapes of gray values by varying the size of small black dots arranged on a regular grid. In summary, I recommend to better position the novelty of the approach with respect to the aforementioned works. By not considering and incorporating prior approaches, the potential capabilities and applicability of APR have been limited. It is especially important to compare its performance to existing approaches. These comparisons could include one similar method (e.g. super-pixels, wavelet), widely used in bioimaging. More generally, some discussion is needed of what, exactly, can and cannot be done with existing tools, to make the novelties and benefits of the proposed representation more explicit. The authors made several efforts in that direction but they did not give illustrations to support all the claims.

2. Another important question to be addressed is related to the exploitation of APR images. The authors clearly demonstrate the theoretical optimality of APR, the robustness of APR to noise, and the capability of APR to save memory. All these items are well presented with many details (see Supplementary Material) in the paper. I'm convinced that APR is appropriate for image visualization, manipulation and image management. Moreover, a large family of image segmentation and classification algorithms can be applied to the adaptive subsampled images. Also, one can benefit from the APR representation to apply more dedicated and powerful Laplacian graph-based image processing approaches (this property could be more emphasized in the paper). While this is fine as such, noise statistics are probably modified when the non-linear APR approach is applied to the raw images. Formally, Gaussian-Poisson noise in raw images is not preserved in the APR images. This means that regularized image restoration and deconvolution methods cannot be applied since the characteristics of noise are not well defined. However, I guess that the novel characteristics of noise in APR images can be empirically analyzed with simulations. The possibility to compare noise statistics in raw and APR images is important to avoid image processing problems and could be more illustrated in the paper. It is worth noting that the authors satisfyingly evaluated the gradient magnitude in raw and APR images (see Supplementary Material). The results tend to suggest that the noise is approximately Gaussian in APR images. Nevertheless, this issue should be addressed, and potential solutions should be included in the next version.

3. In practical imaging, the potential user needs to be instructed about the parameters to compute

APR. The authors should at least discuss this necessity and provide what the prerequisites are for the input.

5. The claim that the proposed APR will overcome the memory bottleneck is true, but the acquired raw data cannot be removed once APR is applied. It is actually mandatory to store the original images in cell imaging (reproducible research) if published. APR appears to be more appropriate for visualization and manipulation.

6. A number of specific comments on the text must be addressed:

– Page 3, I found that the presentation of the “Implied Resolution Function” is not easy to understand in the second Section (APR). Figure 2 is not easy to follow for non-specialists. I recommend to make an effort to express the idea more intuitively. A simple sketch could be added since Figure 2 is quite dense”.

– Pages 3-8 present the properties of APR with details but the related sections are accessible to a limited audience, for example, developers of image processing methods and algorithms.

– Page 10. In Figure 4, the APR reconstruction images are less noisy than the original images. Does it mean that APR implicitly removes noise ? Which amount of noise is removed and what are the statistics of the residual noise in the APR images (see also comments above) ?

– Page 11. The authors claim that the compression ratios are comparable to custom lossy compression methods designed specifically for storing of fluorescence microscopy images. I suggest to show such a compressed image to enlighten the difference between the images and methods.

– Page 13. It is doubtful that the usual pixelwise Markov Random Fields (MRF) methods can be applied to APR images since the regular neighborhood of a given pixel (4 or 8 neighbors) are corrupted by the APR approach. An additional recommendation could be given here. Actually, the super-pixel representation embedded in a MRF framework requires the manipulation of graphs with a variable number of neighbors at each node (gravity center of super-pixels). In Figure 5C, the number of neighbors is not the same for all nodes with APR. Accordingly, the impact of the graph construction should be discussed if pixelwise algorithms (for instance for optical flow computation or non-parametric image registration) are applied to reconstructed APR images. Graph cuts (image segmentation) and Laplacian graph methods are probably more appropriate to APR in general, as demonstrated in page 16.

REMARKS:

The figures and captions are very dense in general.

Reviewer #2 (Remarks to the Author):

The authors propose an adaptive representation of images -the APR- as a replacement to pixels. It aims at providing a more efficient storage and faster/more convenient analysis. The APR is lossy compared to pixels, but that loss is controlled per pixel thus adapted to microscopy and biology. Mathematical proofs and arguments are provided backing the authors claims and efficient algorithms and storage solutions are proposed and implemented. Thorough evaluation of computation of the APR and of its use in analysis is done, both on synthetic and real images.

The problem tackled by the authors is important and urgent. Modern fluorescence microscopy is acquiring data at an ever increasing rate, leading to ever larger images/volumes, longer times and higher resolutions. Traditional file formats are reaching their usability limit and a number of alternative formats have recently been proposed. The authors propose a thorough solution in the form of a new image representation. It is a well defined, fully developed solution designed from the ground up for modern biology uses. Importantly (and not so commonly), a full mathematical development is provided to strengthen their claims and analyse the properties of their construction and algorithms. Applications and validation include practical use cases and benchmarks, showing that the aim is a practical, truly usable solution.

Comments:

- Importantly, for that work to be actually used outside of the authors' lab and become more than 'yet another file format', it needs to be integrated into actual frameworks and pipelines used by the community. At the time of this review, only the C++ library is released, but for a wide adoption bindings in python/matlab/java/ImageJ are needed. They are promised by the authors but should, I think, be released concomitantly to the publication of the article to benefit from the community interest. Additionally bioformats is becoming the de-facto standard for microscopy file format interoperability. Are there plans for a bioformats integration?
- Along the same line, and given that usual algorithms implementation, which would be the one used in practice, are not APR-aware, the authors should show how well would the APR be integrated with, say, a standard ImageJ/matlab/python pipeline: what's the performance of a standard implementation of a linear filter, for example? One interesting test case that could be for example to see the APR reused to recompute a published analysis. If the authors could take one of their previous biological publications/collaborations, reuse the same pipeline of plugins to recompute some quantifications and redo one of its figures using the APR, and find similar results, it would show the compatibility and convince readers less impressed by mathematical arguments that the APR is, at least, just as good as pixels in practical applications
- The APR is from the start developed with very large acquisitions in mind. In the text, the authors show benchmark on data up to 4Gb. That is still up to two orders of magnitude smaller than the very large acquisition that will soon be commonplace. I would like to see some specific comments if not experiments on that in the text: does the APR scale all the way up (to Tb sized images), in terms of computation, storage and uses? In particular is parallel access (read and/or write) possible? What is the speed of arbitrary patch extraction (i.e. reading a particulate rectangular volume from the data)?
- The mathematical arguments are, overall, rigorously written and convincing. However, because of the constraints of the format, where all mathematical arguments are relegated to the supplementary material in an order which is not always natural, they can be frustratingly hard to follow. That is particularly obvious at the start, where the first equation of SuppMat 2 involves ξ_i , which I am not sure are defined anywhere (SuppMat 10 maybe?). I am not sure what the best way would be. A proper, rigorous, in order, mathematical write up would lead to the supplementaries not being in sync with the main text part they are a supplementary to, which would pose other issues...

Overall, this article provides a potentially very useful and rigorously defined solution to a very current problem. However, I think that to fully support its claim of being more than yet another format/image representation, integration into existing scientific workflow to work on the very large images it is claimed to be made for needs to be demonstrated.

Response to Reviewers for Manuscript NCOMMS-18-07550-T - “Forget Pixels: Adaptive Particle Representation of Fluorescence Microscopy Images”

By B. L. Cheeseman, U. Günther, M. Susik, K. Gonciarz, and I. F. Sbalzarini

We thank the reviewers for their thorough reading of the manuscript and the constructive comments that have helped improving the work. We have accounted for all of their comments in the revised version of the paper. For better overview, the revisions are marked in red in the manuscript. In addition, we provide point-by-point replies to the reviewer’s comments here below in *green italics*, quoting the reviews in black print. In cases where multiple points are answered with the same reply, the points have been grouped.

REVIEWER 1:

The paper presents a novel content-adaptive representation (“Adaptive Particle Representation – APR) for 2D-3D fluorescence microscopy image manipulation, visualization, and processing. The potential of APR is demonstrated with theory and experiments on real image data. The paper is well written, and the results are interesting. The properties of APR are somewhat too technical in the first part of the paper. This part is only accessible to specialists in image analysis and image representation.

We thank the reviewer for this appreciation of our work. We have paid particular attention to the readability of the technical sections in the first section. We have endeavored to make the first paragraphs ‘self-contained’ such that a reader not interested in technical details can skip them without compromising understanding the fundamental concepts, but someone interested in reproducing the work finds all information necessary. We feel that further reduction or movement of these new ideas to the supplementary material would reduce the utility of the text as an introduction to, and reference for, the new methods and ideas presented for the first time in this paper.

1. The state of the art is partially covered and the manuscript ignores prior relevant work on similar representations.

We agree with the reviewer that there are many more related works, which we have not mentioned. However, a more in-depth comparison between conceptually related methods has been addressed to the Discussion section.

We do agree that a broader discussion of context, including some details on limitations that do not allow the satisfaction of the representation criteria is required and would be useful in the introduction. In this direction, we have also added a more general overview of related image-based methods and included comments on their limitations, deferring the mentioning of adaptive representations from fields other than image processing to the discussion in order to keep the presentation concise.

The authors focus on super-pixels and wavelet-based image representation but do not explicitly mention recent work about sparsity-based image representation: many microscopy images are composed of fluorescent objects against a dark background as illustrated in the paper and videos.

We agree that with the above-mentioned change of focus on image representations in the introduction, sparsity-based image representations should also be mentioned. We have added the appropriate references and comments to include dictionary-based sparsity methods and discuss their limitations in this context.

Another category of approaches in image representation focused on image level lines according to the Matheron’s theory. The Fast Level Set Transform (Monasse and Guichard, 2000) is a typical algorithm used to decompose the image into a tree of shapes with no loss of

information.

We thank the reviewer for highlighting these works, of which we were not aware. Both are relevant, especially given that the contributions directly focus on accounting for local contrast variations in an image. We have added appropriate comments and references to the introduction and discussion.

Finally, I found that the APR concept can be seen as generalization of the popular halftoning (and dithering) method introduced several decades ago for image printing. The basic idea is to simulate shapes of gray values by varying the size of small black dots arranged on a regular grid.

We agree that the APR, when visualized by its set of underlying particles, bears strong visual similarity with Half-toning or Floyd–Steinberg error-diffusion results. Indeed, the adaptive mesh-based methods of Yang et al. 2013 (ref. 24) uses such a half-toning approach to help accelerate performance. We have addressed this similarity by addition of a sentence in the discussion. However, the exploration of deeper theoretical comparisons between the two methods is beyond the scope of this work and we believe that their mathematical foundations are fundamentally different.

In summary, I recommend to better position the novelty of the approach with respect to the aforementioned works. By not considering and incorporating prior approaches, the potential capabilities and applicability of APR have been limited. It is especially important to compare its performance to existing approaches. These comparisons could include one similar method (e.g. super-pixels, wavelet), widely used in bioimaging.

We agree with the reviewer's comment and have modified the introduction and the conclusions to better review previous image-based adaptive representations and highlight their limitations, as detailed above.

We also agree that comparison with existing techniques, where applicable, is valuable. However, we note that none of the previous techniques simultaneously satisfy the requirements set out in the Representation Criteria, rendering a complete comparison infeasible. For example, although techniques exist for accelerating specific tasks, such as segmentation using super-pixels, or lossy compression methods using wavelets, they do not provide general solutions across processing tasks, nor provide similar theoretical frameworks making the comparisons meaningful. Further, in most circumstances any previous techniques can be used in conjunction with the APR (that is the techniques can also be applied to the APR, just as to a pixel image). This renders direct comparisons, in aid of choosing one solution over another, not appropriate.

Notwithstanding these methodological issues, we have supplemented the text to provide two direct comparisons with existing methods for particular tasks. First, we provide additional 2D benchmarking results in comparison with wavelet thresholding (See SuppMat 12), in addition to the existing theoretical arguments given in the text. Wavelet thresholding has been chosen, due to its known optimality properties (See ref. 31 (DeVore et al. 1992) and 42 (Donoho and ohnstone 1994)), and the established implementations in 2D available through, e.g., Matlab. We note that these results, including the theoretical arguments, show that the error adaptation of the APR, is sub-optimal regarding the number of non-zero coefficients (sparsity) required to obtain a specific error norm. However, this tradeoff comes at the benefit of having direct pointwise error control, the accounting of local contrast changes through the local intensity scale, and the APR providing a representation that is 'closer' to typical pixel representation allowing more direct extension of existing algorithms to the APR. Further, if desired, the wavelet transform can also be performed on an APR. We have slightly adjusted the comments in the discussion to better highlight this.

The second direct comparison we provide is for image compression. Here we apply a recently proposed lossy compression algorithm for large fluorescence images (ref. 31) to both a pixel image and its APR. Here we have re-implemented the technique to allow direct comparison for both pixels and the APR (See SuppMat 20.1), and updated the "Storage Requirement"

section. While we find similar performance in terms of image quality loss for given parameters, the memory compression ratio is significantly better on the APR. These results highlight how, rather than a replacement for existing techniques, in most cases, the APR can instead be used to accelerate them, concerning computational and memory costs, and also potentially improve the results. Therefore, the APR nicely plays with many existing methods and does not aim to replace them, nor to provide a mutually exclusive alternative.

Another direct comparison that would seem to be relevant would be with the super-pixel approaches for segmentation presented in ref. 14. However, as in the above cases, the super-pixel and optical-flow approaches could also take an APR as input. Hence, providing a performance comparison is in line with that already provided for the graph cuts (ref. 35, Boykov et al. 2014) benchmark in the text.

More generally, some discussion is needed of what, exactly, can and cannot be done with existing tools, to make the novelties and benefits of the proposed representation more explicit. The authors made several efforts in that direction but they did not give illustrations to support all the claims.

We believe that the revised manuscript, in particular including the points mentioned above, provides a complete illustration of the claims and better highlights novelty and limitations, also in comparison to many other approaches.

2. Another important question to be addressed is related to the exploitation of APR images. The authors clearly demonstrate the theoretical optimality of APR, the robustness of APR to noise, and the capability of APR to save memory. All these items are well presented with many details (see Supplementary Material) in the paper. I'm convinced that APR is appropriate for image visualization, manipulation and image management. Moreover, a large family of image segmentation and classification algorithms can be applied to the adaptive subsampled images. Also, one can benefit from the APR representation to apply more dedicated and powerful Laplacian graph-based image processing approaches (this property could be more emphasized in the paper).

We thank the reviewer for this suggestion. We now explicitly point this out in the Image Processing Summary section.

While this is fine as such, noise statistics are probably modified when the non-linear APR approach is applied to the raw images. Formally, Gaussian-Poisson noise in raw images is not preserved in the APR images. This means that regularized image restoration and deconvolution methods cannot be applied since the characteristics of noise are not well defined. However, I guess that the novel characteristics of noise in APR images can be empirically analyzed with simulations. The possibility to compare noise statistics in raw and APR images is important to avoid image processing problems and could be more illustrated in the paper. It is worth noting that the authors satisfyingly evaluated the gradient magnitude in raw and APR images (see Supplementary Material). The results tend to suggest that the noise is approximately Gaussian in APR images. Nevertheless, this issue should be addressed, and potential solutions should be included in the next version.

– Page 10. In Figure 4, the APR reconstruction images are less noisy than the original images. Does it mean that APR implicitly removes noise? Which amount of noise is removed and what are the statistics of the residual noise in the APR images (see also comments above)?

We agree with the reviewer that the noise statistics of an image is an important piece of prior knowledge that is typically exploited when designing image-analysis methods. Using an inappropriate noise model has also repeatedly shown to lead to sub-par results, particularly in variational models. We have therefore added two direct studies of how the noise of the pixel image maps into the APR.

First, we have provided additional synthetic image results in SuppMat 7.6, showing the impact of the APR adaptation on the noise distribution of pixels vs. particles. Secondly, we have

highlighted the link with the existing theoretical arguments in SuppMat 7.5, regarding the noise distribution of particles and given explicit theoretical results for the cases of both Gaussian and Poisson noise distributions on the pixels. We note that these results depend on the method chosen to estimate the particle intensity values from the original image, more than on the APR itself. Further, we find that this results in a natural decomposition of the noise distribution by Particle Cell level, which could be another interesting property of the APR to be studied in future work.

Second, we highlight that this level-wise change of the noise distribution requires adaptive regularization terms and noise models in model-based image analysis approaches. We note however that, given the original noise distribution, following analysis as in SuppMat 7.5, a partitioned noise energy term could be formulated level-by-level. This is especially true under the assumption of Gaussian noise. Further, the noise distribution in content-rich areas, notably around edges in the images, is unchanged. For image-analysis methods that focus on these areas, such as segmentation methods, the same noise model/term as on pixels may thus be used. We now briefly highlight this point in the discussion subsection of the processing benchmarks in the main text.

3. In practical imaging, the potential user needs to be instructed about the parameters to compute APR. The authors should at least discuss this necessity and provide what the prerequisites are for the input.

We note that in SuppMat 14, we provide a detailed discussion of the parameters required to be set when using the APR, and we provide suggestions on how to set the values and what they mean. The parameters are also mentioned and referenced in the main text in the "3D Fluorescence APR Implementation" section.

5. The claim that the proposed APR will overcome the memory bottleneck is true, but the acquired raw data cannot be removed once APR is applied. It is actually mandatory to store the original images in cell imaging (reproducible research) if published. APR appears to be more appropriate for visualization and manipulation.

We agree with the reviewer that these regulatory requirements can prevent the discarding of the original pixel images. However, we note that this can be circumnavigated if the APR is formed directly on the fly during acquisition on the microscope, as it would then arguably qualify as the "raw data". A notable precedence for this was provided by the projection-based techniques presented in Schmid et al. 2013 (ref. 25). However, when this is not the case or not desired, the difference image between the original image and the APR can be stored in long-term storage for archival, providing lossless access to the original data at any time while still using significantly less storage. All image analysis and processing could then be carried out on the APR, benefitting from computational savings and leaving the raw data untouched for archiving. We have added a comment to the conclusions on this issue.

6. A number of specific comments on the text must be addressed:

– Page 3, I found that the presentation of the "Implied Resolution Function" is not easy to understand in the second Section (APR). Figure 2 is not easy to follow for non-specialists. I recommend to make an effort to express the idea more intuitively. A simple sketch could be added since Figure 2 is quite dense".

We have adapted the first two introductory paragraphs of "The Adaptive Particle Representation" section to make the introduction of these concepts simpler in hope of making them accessible to a broader audience. We have given explicit attention to the first introduction of the Implied Resolution Function and now believe with the existing figures the explanation should be clearer. We have also removed two sub-figures from Figure 2 to reduce the over-all complexity of the figure and focusing on novel concepts.

From discussions with colleagues regarding this figure, they have found the technical details to be of great utility for the understanding of the APR and for helping them program or

implement the APR in their codes. Hence, we believe further reduction will limit the usability of the paper as a reference for future users.

– Pages 3-8 present the properties of APR with details but the related sections are accessible to a limited audience, for example, developers of image processing methods and algorithms.

We agree that this section does provide technical details regarding the APR, and we have modified the first two paragraphs of this section to provide a clearer and more straightforward overview of the APR. Further, we included a sentence at the beginning of the section to guide the reader for whom these details may not be of relevance.

Indeed, these sections require a certain degree of technical aptitude and background to be engaged with correctly. However, we believe further reduction and simplification would result in misrepresentation of our ideas and concepts in upcoming software codes. Also, this would reduce the potential impact of the ideas being able to be extended outside the context of fluorescence imaging. We do agree that the degree of technicality makes this section un-accessible for some readers, though. However, using the APR does not require understanding the technical details of it, nor does it preclude appreciation of the underlying concepts. We therefore wish to keep the section at its present, revised level of detail.

– Page 11. The authors claim that the compression ratios are comparable to custom lossy compression methods designed specifically for storing of fluorescence microscopy images. I suggest to show such a compressed image to enlighten the difference between the images and methods.

We agree with the reviewer and we apologize for this omission. We have extended the direct analysis and comparison of the APR by also including the recent “within noise level” compression method (Balazs et al. 2017 ref. 30) in SuppMat 20.1, which is the state of the art in the field of lossy microscopy image compression. We show a comparison between an original image, the APR, the lossy compressed APR, and the lossy compressed pixel image in SFigure 38. We have also changed the main-text comments to reflect the complementary nature of the APR and existing lossy techniques. The results from the equivalent pixel algorithm have also been added for all exemplar datasets in Table 4.

– Page 13. It is doubtful that the usual pixelwise Markov Random Fields (MRF) methods can be applied to APR images since the regular neighborhood of a given pixel (4 or 8 neighbors) are corrupted by the APR approach. An additional recommendation could be given here. Actually, the super-pixel representation embedded in a MRF framework requires the manipulation of graphs with a variable number of neighbors at each node (gravity center of super-pixels). In Figure 5C, the number of neighbors is not the same for all nodes with APR. Accordingly, the impact of the graph construction should be discussed if pixelwise algorithms (for instance for optical flow computation or non-parametric image registration) are applied to reconstructed APR images. Graph cuts (image segmentation) and Laplacian graph methods are probably more appropriate to APR in general, as demonstrated in page 16.

The reviewer is correct regarding the APR Particle graph’s anisotropic and locally varying neighborhood. This indeed renders the APR Particle graph inappropriate for techniques that require a regular isotropic neighborhood. However, in such cases, a locally isotropic neighborhood patch can always be constructed on the fly around each particle. This local patch reconstruction is similar to the approaches used for the pixel-based filtering on the APR. We thank the reviewer for highlighting this additional possibility, and we now explicitly mention this in the segmentation section, where we added an additional remark about how to use regular-neighborhood methods on the APR.

REMARKS:

The figures and captions are very dense in general.

We made an extra effort to simplify the captions without losing information. We have also simplified Figure 2, and the central Figure 3, moving panel A to the supplement. From

feedback, we have found pipeline flowchart to be effective, however, the schematic of the separability property, although useful, is likely too technical to justify such real-estate in the main text.

REVIEWER 2:

The authors propose an adaptive representation of images -the APR- as a replacement to pixels. It aims at providing a more efficient storage and faster/more convenient analysis. The APR is lossy compared to pixels, but that loss is controlled per pixel thus adapted to microscopy and biology. Mathematical proofs and arguments are provided backing the authors claims and efficient algorithms and storage solutions are proposed and implemented. Thorough evaluation of computation of the APR and of its use in analysis is done, both on synthetic and real images.

The problem tackled by the authors is important and urgent. Modern fluorescence microscopy is acquiring data at an ever increasing rate, leading to ever larger images/volumes, longer times and higher resolutions. Traditional file formats are reaching their usability limit and a number of alternative format have recently been proposed. The authors propose a thorough solution in the form of a new image representation. It is a well defined, fully developed solution designed from the ground up for modern biology uses. Importantly (and not so commonly), a full mathematical development is provided to strengthen their claims and analyse the properties of their construction and algorithms. Applications and validation include practical uses case and benchmarks, showing that the aim is a practical, truly usable solution.

We thank the reviewer for this appreciation of our work. We hope the revisions have further improved it.

Comments:

- Importantly, for that work to be actually used outside of the authors lab and become more than 'yet another file format', it needs to be integrated into actual frameworks and pipelines used by the community. At the time of this review, only the C++ library is released, but for a wide adoption, bindings in python/matlab/java/ImageJ are needed. They are promised by the authors but should, I think, be released concomitantly to the publication of the article to benefit from the community interest. Additionally, bioformat is becoming the de-facto standard for microscopy file format interoperability. Are there plans for a bioformat integration?

We entirely agree with the reviewer, and a concerted effort is now underway to provide integration that can make the APR more widely available and useful.

Importantly, a uniform-resolution pixel image can always be reconstructed from the APR on the fly at the expense of the additional memory (but not storage), and the C++ library already allows this. This already enables direct use of non-APR-aware algorithms without modification. However, the computational cost of these algorithms will remain unchanged from the pixel case. The ideal case, however, is when algorithms can be adapted to fully benefit from the APR and the additional information it provides. This will be future work.

- Along the same line, and given that usual algorithms implementation, which would be the one used in practice, are not APR-aware, the authors should show how well would the APR be integrated with, say, a standard ImageJ/matlab/python pipeline: what's the performance of a standard implementation of a linear filter, for example?

We are currently developing an interface between the APR C++ Library and Fiji/ImageJ that will allow interoperability between existing pixel algorithms and algorithms directly on the APR. This is planned to include the integration into existing tools such as BigDataViewer for by-slice visualization without reconstructing the original image. The interface will also allow any image that can be read by Fiji (through BioFormats) to then be used with LibAPR without intermediate conversion to TIFF, as currently is required. Such interface will enable hybrid pipelines where some steps could use the pixel image, and some the APR. The C++ Library can also already be used, via wrappers, from both Java and Python. Java wrappers also allow the potential calling of functions from within Matlab.

One interesting test case that could be for example to see the APR reused to recompute a published analysis. If the authors could take one of their previous biological publication/collaboration, reuse the same pipeline of plugins to recompute some quantifications and redo one of its figure using the APR, and find similar results, it would show the compatibility and convince readers less impressed by mathematical arguments that the APR is, at least, just as good as pixels in practical applications

We agree with the reviewer that such a comparison would make a convincing use case for adopting the APR in a particular task or workflow. However, we believe such an example would be too specific, and prone to overfitting. We therefore believe that such specific comparisons are better left to future work, pipelines, and presentations. Instead, we here wish to focus on the general concepts and benchmark of the APR that are independent of a specific biological application.

In a more general context, the Reconstruction Condition provides direct control over the point-wise representation accuracy. Hence the accuracy of the pipeline is implementation and algorithm dependent. We have attempted to highlight these potential benefits and pitfalls with the processing examples given in the last section of the paper.

- The APR is from the start developed with very large acquisitions in mind. In the text, the authors show benchmark on data up to 4Gb. That is still up to two orders of magnitude smaller than the very large acquisition that will soon be commonplace. I would like to see some specific comments if not experiments on that in the text: does the APR scale all the way up (to Tb sized images), in term of computation, storage and uses? In particular is parallel access (read and/or write) possible? What is the speed of arbitrary patch extraction (i.e. reading a particulate rectangular volume from the data)?

It appears that we mistakenly omitted a reference to the supplementary material that provides exactly these details in SFigure 35. We apologize for this. In this benchmark, we show scaling on images up to 100GB. We have updated the main text to now highlight this. Given enough computer memory, one can go to arbitrarily large images. We have also highlighted this using a test case image of size 320 GB and provide the comparative results.

In addition, the separability property of the APR allows a block-wise computation strategy, which will allow processing of very large images, beyond the size of available memory on the machine. However, this method development and associated software is still underway and will be subject of future work and publications.

- The mathematical arguments are, overall, rigorously written and convincing. However, because of the constrains of the format, where all mathematical arguments are relegated to the supplementary material in an order which is not always natural, they can be frustratingly hard to follow. That is particularly obvious at the start, where the first equation of SuppMat 2 involve ξ_i , which I am not sure are defined anywhere (SuppMat 10 maybe?). I am not sure what the best way would be. A proper, rigorous, in order, mathematical write up would lead to the supplementaries not being in sync with the main text part they are a supplementayry to, which would pose other issues...

We apologize for this omission. The undefined symbols were a mistake, as a subsection was shortened during editing. This has been remedied, and we thank the reviewer for noting this. We have also streamlined the supplement, by inclusion of a table of contents. We believe, this aids in readability and access and serves as an index to the large volume of material we provide.

Overall, this article provide a potentially very useful and rigorously defined solution to a very current problem. However, I think that to fully support it's claim of being more than yet another format/image representation, integration into existing scientific workflow to work on the very large images it is claimed to be made for needs to be demonstrated.

We agree that realizing the full potential of the APR requires continued development and integration with existing tools and pipelines. Such efforts do take time and resources but are

currently under-way as the APR is becoming adopted by the open-source community.

REVIEWERS' COMMENTS:

Reviewer #1 (Remarks to the Author):

The authors satisfyingly answered to all my concerns.

They paid attention to all items and accurately provided mathematics, figures and experiments to support several claims.

They made a large effort in general to improve the manuscript and to justify the positioning.

For all these reasons, I consider that the paper can be accepted.

Reviewer #2 (Remarks to the Author):

Concerning specifically the answers of the authors to my comments:

- As of this writing, a python API does not seem to appear in the github repository, where it is still noted as coming soon. Since the authors say in their answer that it is available, maybe it is just not publicly available yet?
- The revised paper indeed show example up to 320Gb, but the stated limitation of having 2.7 time the image size in RAM means that doing so need more than 800Gb of RAM, which is not common place and that going significantly higher than that will be problematic.

I will admit that those comments are partly selfish, as I work with large images in python, and thus may not be able to test/use the APR yet... The size limitation though is really problematic for an algorithm which, from the start, is aimed at large datasets; are there ways to circumvent it, or improvements being developed that would alleviate it?

Apart from those comments, the revised manuscript is significantly improved and the work is overall of high relevance and quality. Thus I would recommend publication, provided the release of the python API.

Response to Reviewers for Manuscript NCOMMS-18-07550-T - “Adaptive Particle Representation of Fluorescence Microscopy Images”

By B. L. Cheeseman, U. Günther, K. Gonciarz, M. Susik, and I. F. Sbalzarini

We thank the reviewers again for their thorough reading of the manuscript and the constructive comments that have helped improving the work. We are pleased they find the work suitable for acceptance. We provide point-by-point replies to the reviewer’s comments here below in *green italics*, quoting the reviews in black print.

Response to reviewer 2:

- As of this writing, a python API does not seem to appear in the github repository, where it is still noted as coming soon. Since the authors say in their answer that it is available, maybe it is just not publicly available yet?

We are pleased regarding the reviewer's interest in the Python wrappers. We apologize for the confusion, the basic python wrappers were available through the developPython branch of the GitHub repository at <https://github.com/cheesema/LibAPR/tree/developPython>. However, we have now included these in the main master branch, which makes it easier to find them. We note that the functionality allows generation of the APR and reconstruction of images.

If the reviewer requires further assistance or is interested in additional functionality, we would be happy to help in any way. By contacting either the first or last authors directly following the acceptance of the manuscript.

- The revised paper indeed show example up to 320Gb, but the stated limitation of having 2.7 time the image size in RAM means that doing so need more than 800Gb of RAM, which is not common place and that going significantly higher than that will be problematic.

Yes, the reviewer is correct that the current pipelines requirement for 2.7 times memory, although not a large requirement in usual contexts, when used for extremely large (>20GB) images this results in large amounts of RAM being required, making the formation of the APR not possible without dedicated hardware for these datasets. We are aware that not all researchers have access to servers with 1TB of RAM as used for the presented benchmarks. For this use-case, we agree this is a limitation, and that as the reviewer points out, such large datasets present an ideal use-case for the APR.

Addressing this issue is the subject of on-going research. For these use-cases the APR can be formed using a block-wise decomposition, requiring significant adjustments to the algorithms, and additional software development. We are now explicitly mentioning this in the Conclusions section. These changes, allow then the formation of the same APR, with memory requirements only restricted to 2.7 times the local block used. We feel that these changes are outside the scope of the research paper presented here. However, we note that preliminary work has been done, and we will make this available as soon as possible.